# Light-programmable mechanical computing via polyaniline composite film

Xiunan Yan [1,6], Yixiang Li [1,6], Yichen Zhao [1,6], Chen Pan [2] ✉,
Shengnan Yan[1], Dong Yang[2], Gong-Jie Ruan [1], Hang Zhao[1], Fanqing Chen [1],
Xing-Jian Yangdong [1], Pengfei Wang [1], Wentao Yu [2], Yuekun Yang [1,3],
Cong Wang [1], Bin Cheng [2], Shi-Jun Liang [1,4,5] ✉ & Feng Miao [1,5] ✉

Mechanical computing represents a highly promising paradigm for environment-adaptive information processing. However, existing implementations are generally constrained by limited architectural scalability, and their modes of application in practical scenarios remain insufficiently defined. Here, we develop a light-programmable mechanical computing system that not only performs scalable logic operations but also enables environment-adaptive optical camouflage. The system is based on a polyaniline composite film (PCF) that integrates light-responsive expansion–contraction elements with a flexible conductive layer. Light illumination dynamically modulates the conductive pathways, giving rise to optically controlled single-pole single-throw (SPST) and single-pole double-throw (SPDT) relays that reconfigure signal transmission routes. Interconnecting these relays enables the construction of basic logic gates and 2-bit full-adder circuits, establishing a scalable paradigm for light-programmable mechanical computation. Moreover, we implement an adaptive camouflage function that senses environmental textures and generates matching optical patterns, demonstrating potential for intelligent skin applications capable of environmental interaction. This work establishes a light-programmable, pathway-reconfigurable mechanical computing framework, expanding possibilities for autonomous and adaptive intelligent systems.

Mechanical computing has recently gained increasing attention for its potential to operate intelligently in dynamically changing environments[1–7]. In contrast to conventional electronics-based architectures, mechanical computing systems offer a decentralized, electronics-free approach to information processing[8–10], which is inherently immune to electromagnetic interference and can function reliably under extreme conditions, such as high temperature, radiation, and corrosion[11–13]. This emerging concept of mechanical intelligence represents a non-von Neumann computing paradigm in which sensing, actuation, and computation are co-localized within the material itself[14–17], enabling low-latency and low-power responses to external stimuli[18–26].

Recent advances in mechanical computing have largely focused on controlling material deformation to encode and process

[1]Institute of Brain-Inspired Intelligence, National Laboratory of Solid State Microstructures, School of Physics, Collaborative Innovation Center of Advanced Microstructures, Jiangsu Physical Science Research Center, Nanjing University, Nanjing, China. [2]Institute of Interdisciplinary Physical Sciences, School of Physics, Nanjing University of Science and Technology, Nanjing, China. [3]School of Intelligence Science and Technology, Nanjing University, Suzhou, China. [4]Chemistry and Biomedicine Innovation Center (ChemBIC), Nanjing University, Nanjing, China. [5]Institute of Brain-Machine Interface, Nanjing University, Nanjing, China. [6]These authors contributed equally: Xiunan Yan, Yixiang Li, Yichen Zhao. ✉e-mail: chenpan@njust.edu.cn; sjliang@nju.edu.cn; miao@nju.edu.cn

information[13,20,27–57]. Mechanical deformation can be directly induced by applied forces, driving mechano-electro-coupling materials to produce electrical or optical outputs for computation[13,27,39–45]. Besides, thermal stimuli have been utilized to trigger deformation-based logic operations in thermally driven mechanical computing systems[46–48]. Moreover, integrating deformation mechanisms responsive to electrical or magnetic fields with circuit design has expanded the possibilities for multifunctional and reconfigurable mechanical computation[28,49–54]. Collectively, these advances highlight that controlling deformation through multiple physical fields is central to advancing mechanically based computation.

Despite this progress, two key challenges persist. First, most existing mechanical computing systems remain limited in computational scalability, as their architectures are typically confined to single-device or few-device demonstrations without an effective scheme for interconnection or expansion of computing capability. Second, the practical application modes of these systems remain insufficiently explored, and how they can leverage environmental stimuli to perform context-specific information processing and achieve functions of practical value has yet to be clearly established. Addressing these limitations requires an alternative class of mechanical computing systems that can be programmed by external stimuli to reconfigure their information processing functions and autonomously perform different tasks in response to dynamic environmental inputs. In this work, we develop a light-programmable mechanical computing system that simultaneously achieves scalable logic operations and environment-adaptive optical camouflage. The system is built upon a polyaniline composite film (PCF) that integrates multiple responsive components into a single deformable platform. Upon illumination, the PCF undergoes optically induced mechanical deformation, dynamically modulating conductive pathways to realize optically controlled single-pole single-throw (SPST) and single-pole double-throw (SPDT) relays capable of reconfiguring signal transmission routes. Building

upon this mechanism, we realized not only universal logic gate operations but also more complex combinational logic functions, including 1-bit and 2-bit full adders, thereby demonstrating the scalability of the mechanical computing architecture. Furthermore, we developed an adaptive camouflage module capable of sensing environmental texture features and generating corresponding optical patterns, thereby demonstrating the potential of intelligent skins based on this module for interactive environmental adaptation. This work establishes a light-programmable, pathway-reconfigurable computing framework, addressing key limitations of mechanical computing in both computational scalability and offering a feasible pathway toward practical implementation in real-world scenarios.

## Results

### The construction of polyaniline composite film for implementing light-programmable mechanical computing

Figure 1 illustrates the structure of the polyaniline composite film (PCF) and the overall strategy for realizing light-programmable mechanical computing. As shown in Fig. 1a, we fabricated a hierarchically coupled composite composed of three functional layers: a polyaniline-poly(N-isopropylacrylamide) (PANi-PNIPAm) layer, a silver nanowires (Ag NWs) conductive layer, and a polydimethylsiloxane (PDMS) layer (molecular structures in Supplementary Fig. 1, material properties detailed in Supplementary Table 1, synthesis procedures in Methods). By integrating these well-established materials[58,59] into a multilayer composite structure, we created an ideal platform that converts optical stimuli into mechanically reconfigurable conductive pathways.

The light-programmable behavior of the designed PCF originates from a hierarchically coupled photo-thermal-mechanical mechanism. Under optical excitation, the PANi component converts incident light into heat via photothermal effects[58], while the PNIPAm component undergoes thermally induced shrinkage once the temperature exceeds

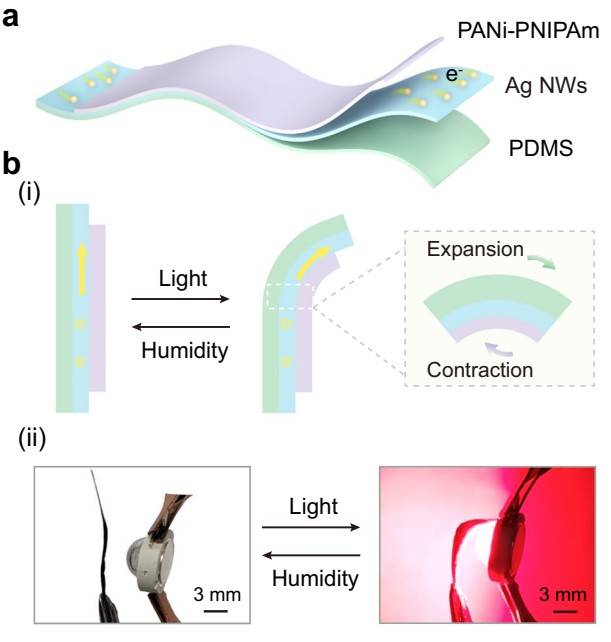

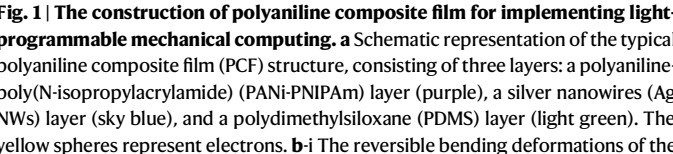

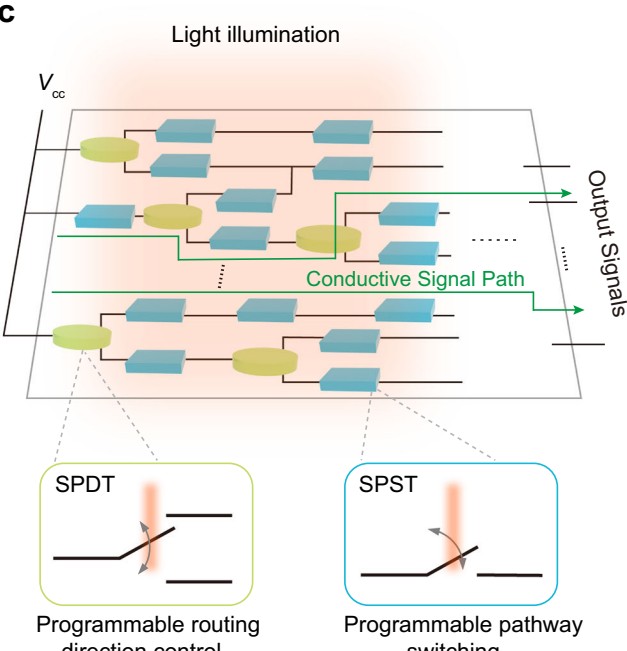

**Fig. 1 | The construction of polyaniline composite film for implementing light-programmable mechanical computing. a** Schematic representation of the typical polyaniline composite film (PCF) structure, consisting of three layers: a polyaniline-poly(N-isopropylacrylamide) (PANi-PNIPAm) layer (purple), a silver nanowires (Ag NWs) layer (sky blue), and a polydimethylsiloxane (PDMS) layer (light green). The yellow spheres represent electrons. **b-i** The reversible bending deformations of the

PCF, induced by variations in light and humidity, result in the deformation of the conductive pathways. ii The corresponding optical images of the above process. Scale bar: 3 mm. **c** The operating mechanism of light-programmable mechanical computing. The terms SPST and SPDT refer to single-pole single-throw and single-pole double-throw relays, respectively. The yellow-green cylinders represent the SPDT, while the blue cuboids represent the SPST.

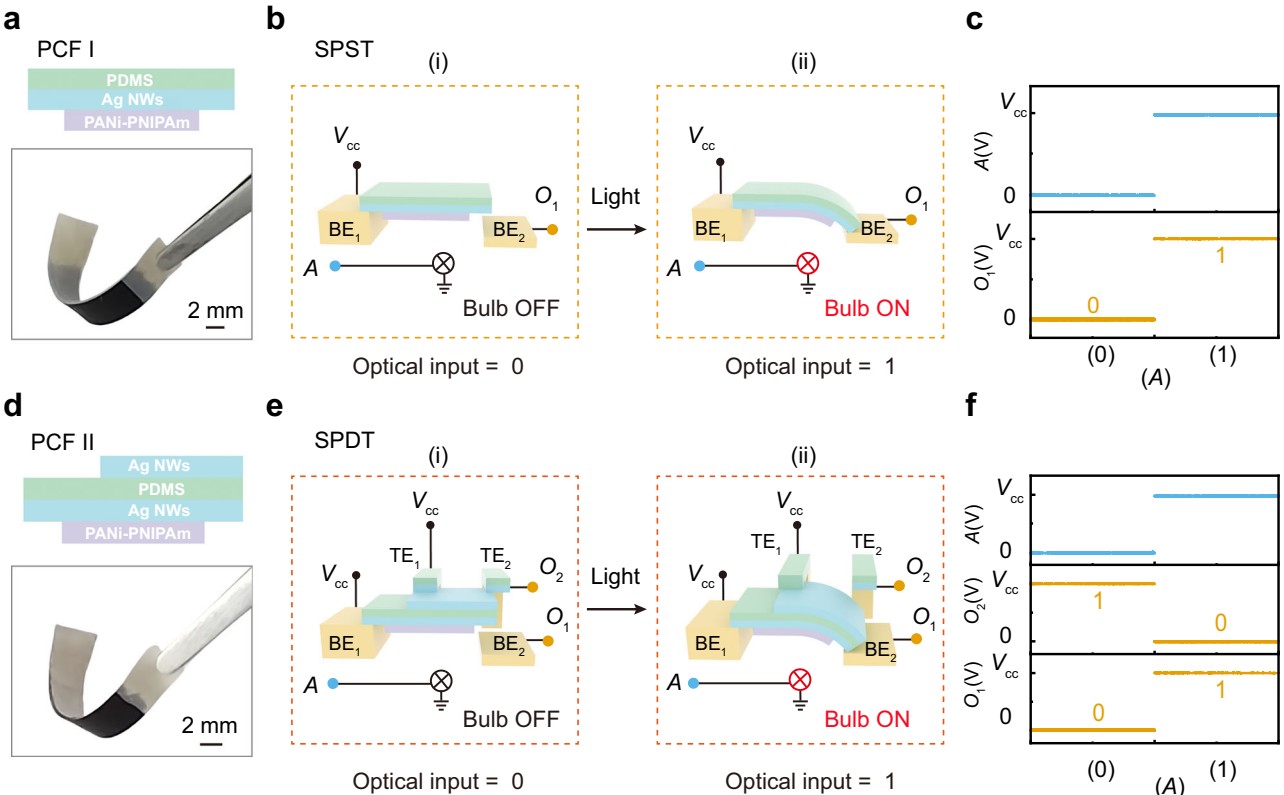

**Fig. 2 | The conductive switching performance of relays. a** The structure and optical image of the polyaniline composite film (PCF) I (PDMS/Ag NWs/PANi-PNI-PAm), with the optical image demonstrating its flexibility. Scale bar: 2 mm. **b** Architecture of the SPST relay. Optical inputs *A* indicated in sky blue solid circles, and the electrical output ($O_1$) in orange solid circles, and the input voltage ($V_{cc}$) in black solid circles. The input light intensity signals control the formation of an electrical pathway, regulating the output signal. **c** Operational state of the SPST relay. **d** The structure and optical image of the PCF II (Ag NWs /PDMS/Ag NWs/PANi-PNIPAm), with the optical image demonstrating its flexibility. Scale bar: 2 mm. **e** Architecture of the SPDT relay. Optical inputs *A* are indicated in sky blue solid circles, and the electrical output ($O_1$, $O_2$) are shown in orange solid circles, and the input voltage ($V_{cc}$) in black solid circles. **f** Operational states of the SPDT relay. The BE and TE represent the bottom electrode and top electrode, respectively.

its lower critical solution temperature (LCST)[60]. The generated heat is conducted by the Ag NWs layer to the underlying PDMS layer, which exhibits positive thermal expansion (SEM characterization shown in Supplementary Fig. 2)[61,62]. The simultaneous contraction of the PANi-PNIPAm layer and expansion of the PDMS layer drive an out-of-plane bending deformation of the film, as shown in Fig. 1b-i and Supplementary Movie 1. Although the Ag NWs layer does not exhibit thermal expansion, its intrinsic flexibility allows it to maintain favorable electrical conductivity throughout the deformation process[63]. As a result, the synthesized PCF can dynamically reconstruct its conductive pathways by modulating its mechanical configuration under light illumination. Detailed experimental results of this light-thermal-mechanical-electrical transduction characteristics are provided in Supplementary Figs. 3 and 4. We further demonstrated that the light-induced deformation of the PCF is fully reversible by adjusting the ambient humidity (Supplementary Movie 2, and the mechanism was described in Supplementary Note 1). As shown in Fig. 1b-ii, the film bends upon illumination and recovers its initial state when the humidity is increased. This reversible actuation endows the PCF with dynamically reconfigurable conductive pathways and confirms that the material can repeatedly transition between distinct mechanical states without fatigue.

Building on this light-responsive modulation of conductive pathways, we constructed light-programmable mechanical computing circuits. Following Shannon's relay-circuit framework[64], we designed two types of PCF-based relays to realize mechanical computation, as illustrated in Fig. 1c. The single-pole single-throw (SPST) relay functions as a binary switch that connects or disconnects a transmission path, serving as the fundamental unit for controlling signal on/off states. The single-pole double-throw (SPDT) relay directs the input voltage ($V_{cc}$) between two alternative output channels, enabling signal routing. By integrating SPST and SPDT relays, we developed a compact and versatile computing architecture in which light intensity dynamically reconfigures the conductive network, allowing the relays to collectively govern both the switching state and the routing of information transmission pathways. This light-induced pathway reconfiguration establishes a promising paradigm for optically controlled mechanical computing, extending traditional actuator-based logic from isolated mechanical motions[65,66] to programmable and network-level information processing.

**Construction of relays with light-programmable characteristics**
We constructed two types of relays based on the polyaniline composite film (PCF) to realize the core physical components of the light-programmable mechanical computing concept. As shown in Fig. 2a, we fabricated a SPST relay using a type-I PCF structure (PCF I: PDMS/Ag NWs/PANi-PNIPAm). This configuration consists of a single silver nanowires (Ag NWs) conductive layer sandwiched between a poly-dimethylsiloxane (PDMS) substrate and a polyaniline-poly(N-iso-propylacrylamide) (PANi-PNIPAm) active layer, forming an isolated conductive channel. The operational mechanism is schematically shown in Fig. 2b. We applied a supply voltage ($V_{cc}$) to bottom electrode 1 ($BE_1$), used light intensity at point *A* as the optical control input, and collected the output signal ($O_1$) from bottom electrode 2 ($BE_2$). Logical inputs were encoded using light stimuli, where illuminated and dark states corresponded to logic 1 and logic 0, respectively. When the light

input was set to 0 (Fig. 2b-i), the PCF remained flat and suspended above $BE_2$, thereby maintaining an open circuit between $BE_1$ and $BE_2$ and yielding an output of $O_1 = 0$, as illustrated in Fig. 2c. Upon switching the light input to 1 (Fig. 2b-ii), light actuation induced a downward bending of the PCF, bringing it into contact with $BE_2$ and closing the circuit. This transition resulted in an output state of $O_1 = 1$, as also shown in Fig. 2c. The logical operation of the SPST relay and its reset mechanism under varying humidity conditions are demonstrated in Supplementary Movie 3. Through this experiment, we achieved light-programmable on/off switching of a single conductive pathway, thus realizing the SPST relay function.

To extend binary switching toward directional signal routing, we designed and fabricated an SPDT relay based on an enhanced type-II PCF structure (PCF II: Ag NWs/PDMS/Ag NWs/PANi-PNIPAm), as illustrated in Fig. 2d. PCF II incorporates an additional Ag NWs layer atop the PDMS substrate, forming dual conductive channels for bidirectional control. As shown in Fig. 2e, we constructed the SPDT relay with two top electrodes ($TE_1$, $TE_2$) and two bottom electrodes ($BE_1$, $BE_2$). The top electrodes were fabricated using a PDMS/Ag NWs composite (Supplementary Fig. 5), which offered high conductivity and mechanical compliance. This flexible composite electrode deformed synchronously with the PCF during actuation, maintaining stable contact and minimizing interfacial stress or delamination that typically occurs with rigid metal contacts. Such mechanical-electrical compatibility ensured robust performance over repeated actuation-recovery cycles. In operation, we applied $V_{cc}$ to $TE_1$ and $BE_1$ and collected the output signals $O_1$ and $O_2$ from $BE_2$ and $TE_2$, respectively. When the light input was 0 (Fig. 2e-i), PCF II remained flat and contacted only $TE_2$, thereby activating the upper conductive path ($O_2 = 1$) while keeping the lower path open ($O_1 = 0$), as illustrated in Fig. 2f. Upon illumination ($A = 1$), the film underwent downward bending (Fig. 2e-ii), disengaging from $TE_2$ and establishing contact with $BE_2$. This transition activated the lower conductive path ($O_1 = 1$) and simultaneously deactivated the upper path ($O_2 = 0$). These results demonstrate that the SPDT relay achieved bidirectional signal routing and light-controlled path selection.

Through systematic experimental characterization, we demonstrated that the switching dynamics of the relays can be effectively tuned by both illumination intensity and environmental conditions (Supplementary Fig. 6). The switching time is defined as the sum of the response time and the recovery time. Under an illumination intensity of 130 mW/cm² at 29 ± 1 °C and 83 ± 3% relative humidity, the single-pole single-throw (SPST) relay exhibited a response time of 1.02 s and a recovery time of 2.07 s, corresponding to a total switching time of 3.09 s and an operating frequency of approximately 0.32 Hz. Similarly, the single-pole double-throw (SPDT) relay showed a response time of 1.20 s and a recovery time of 2.05 s, resulting in a switching time of 3.25 s and an operating frequency of about 0.31 Hz (Supplementary Fig. 7). Notably, increasing the illumination intensity to 300 mW/cm² under the same temperature and humidity conditions significantly reduced the SPST response time to ~0.13 s, while the recovery time remained at 2.4 s. This led to a shorter total switching time of 2.53 s and a correspondingly higher operating frequency of ~0.40 Hz, demonstrating that faster switching can be achieved through optical optimization (Supplementary Fig. 7c). Although the typical switching time remains on the order of seconds, such timescales are well suited for applications including adaptive camouflage, environmental sensing, soft robotics, and low-frequency logic, where reversibility, adaptability, and energy efficiency are more critical than ultrafast response.

The relays also exhibited robust mechanical reversibility and long-term operational stability. To evaluate their robustness under varying humidity, we measured the response and recovery characteristics of the SPST relay (Supplementary Fig. 8a–c). As humidity increased, the response time showed a slight delay, whereas the recovery time was markedly accelerated, confirming stable and reliable recovery

behavior across all humidity levels. Moreover, the SPST relay maintained consistent performance over more than 600 continuous actuation–reset cycles without observable fatigue or degradation (Supplementary Fig. 8d). These results collectively demonstrate the high reliability and robustness of the relays, establishing them as ideal building blocks for constructing light-programmable mechanical computing systems.

To further elucidate the underlying mechanism, we quantitatively analyzed the coupled photo-thermal-mechanical-electrical processes based on a thermal conduction model and beam deformation theory (Supplementary Note 2, Supplementary Figs. 9 and 10). The analysis reveals how light-induced photothermal conversion propagates through heat conduction to actuate mechanical deformation, thereby modulating the conductive pathways and enabling controllable signal transmission. These insights provide theoretical guidance for the design and optimization of light-programmable mechanical computing architectures.

## Realization of scalable logic computing

We constructed scalable logic computing circuits by integrating light-programmable SPST and SPDT relays, as shown in Fig. 3. The SPST relay, capable of controlling the on/off state of a single conductive pathway, serves as the fundamental element for basic Boolean operations. As shown in Fig. 3a-i, we fabricated an AND gate by connecting two SPST relays in series (Supplementary Fig. 11), and the corresponding optical image is presented in Fig. 3a-ii. The circuits were fabricated on flexible styrene-ethylene-butylene-styrene (SEBS) substrates, with the fabrication process and testing setup illustrated in Supplementary Note 3 (Supplementary Figs. 12–14). We measured the output response under different input combinations. When both logic inputs $A$ and $B$ were set to 0, both relays remained open, interrupting voltage transmission from the supply ($V_{cc}$) to the output and resulting in a low output signal $Q_{AND}$ ($Q_{AND} = 0$), as shown in Fig. 3d. When only one input was set to 1, the corresponding relay closed while the other remained open, leaving the circuit incomplete and keeping $Q_{AND}$ at a low level ($Q_{AND} = 0$). Only when both inputs were set to 1 did both relays close simultaneously, completing the circuit and producing a high-level output $Q_{AND}$ ($Q_{AND} = 1$). This behavior confirms the correct implementation of the AND logic operation (Supplementary Fig. 15, Supplementary Movies 4 and 5). We also constructed an OR gate by connecting two SPST relays in parallel, as shown in Fig. 3b-i, and obtained the corresponding optical image shown in Fig. 3b-ii. In this configuration, current conduction occurred when either relay was closed, so the output $Q_{OR}$ remained low ($Q_{OR} = 0$) only when both $A$ and $B$ were set to 0. For all other input combinations, the output $Q_{OR}$ remained high ($Q_{OR} = 1$), consistent with OR logic behavior (Fig. 3d).

To achieve more complex combinational logic, we implemented SPDT relays to perform bidirectional signal routing. As illustrated in Fig. 3c-i, we constructed an XOR gate by connecting two SPDT relays in parallel (Supplementary Fig. 16), and the optical image of the fabricated circuit is shown in Fig. 3c-ii. We measured the output response under all input combinations and obtained the results summarized in Fig. 3d. When both $A$ and $B$ were 0 or both 1, the corresponding conductive paths were simultaneously connected or disconnected, blocking $V_{cc}$ from reaching the output and yielding a low $Q_{XOR}$ ($Q_{XOR} = 0$). In contrast, when $A$ and $B$ were in opposite logic states, one relay conducted while the other remained open, allowing $V_{cc}$ to reach the output and generating a high $Q_{XOR}$ ($Q_{XOR} = 1$) (Supplementary Fig. 17 and Supplementary Movie 6). The detailed operation mechanism of the XOR logic is presented in Supplementary Fig. 18. Furthermore, we realized an inverter (NOT gate) by slightly modifying the XOR configuration, where the lower PCF film was fixed to $BE_1$ and $BE_2$ with silver paste, and only input $A$ was varied. The resulting output corresponded to the logical NOT operation of $A$ (Supplementary Fig. 19).

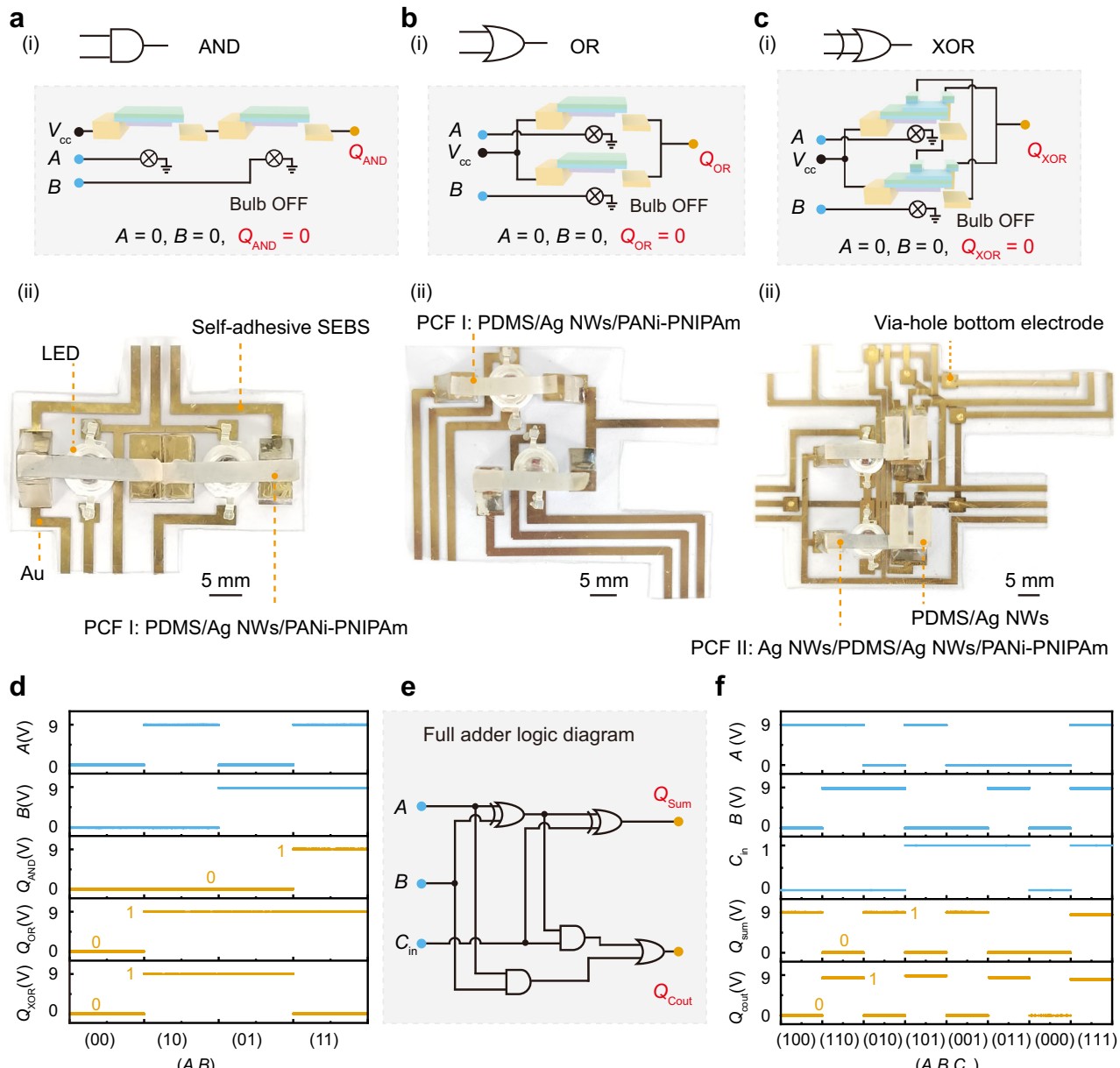

**Fig. 3 | Realization of scalable logic computing. a**-i AND gate circuit constructed with two single-pole single-throw (SPST) relays in series, with optical inputs ($A$, $B$) indicated in sky blue solid circles, electrical output ($Q_{AND}$) in orange solid circles, and the input voltage ($V_{cc}$) in black solid circles. ii The corresponding optical image of the AND gate. Scale bar: 5 mm. **b**-i OR gate circuit constructed with two SPST relays in parallel, with optical inputs ($A$, $B$) indicated in sky blue solid circles, electrical output ($Q_{OR}$) in orange solid circles, and the input voltage ($V_{cc}$) in larger black solid circles. ii The corresponding optical image of the OR gate. Scale bar: 5 mm. **c**-i XOR gate circuit constructed with two single-pole double-throw (SPDT) relays in parallel, with optical inputs ($A$, $B$) indicated in sky blue solid circles, electrical output ($Q_{XOR}$) in orange solid circles, and the input voltage ($V_{cc}$) in larger black solid circles. ii The corresponding optical image of the XOR gate. Scale bar: 5 mm. **d** Measured input and output voltages for all input combinations of ($A$, $B$) for the AND, OR, and XOR gate. **e** One-bit full adder circuit constructed by cascading AND, OR, and XOR gates. Optical inputs ($A$, $B$, $C_{in}$) are indicated in sky blue solid circles, while electrical outputs ($Q_{Sum}$, $Q_{Cout}$) are marked in orange solid circles. **f** Summary of output voltages for various input combinations ($A$, $B$, $C_{in}$) of the full adder, demonstrating scalable and complete logic computing capabilities.

We further demonstrated the scalability of the light-programmable mechanical computing system by cascading SPST and SPDT relays to construct combinational logic circuits. As shown in Fig. 3e, we implemented one-bit full adder by integrating basic logic gates (see detailed assembly method in Supplementary Note 3 and Supplementary Movie 7), and the optical images of the fabricated circuits are provided in Supplementary Fig. 20. The cascading operation was achieved by using the output voltage of one logic gate to drive a light source, which optically triggered the subsequent stage. The one-bit full adder comprises three logic inputs: $A$, $B$, and $C_{in}$, which correspond to two binary values and a carry input. The circuit

generates two outputs: $Q_{Sum}$ and $Q_{Cout}$, representing the least significant bit and the carry of the binary addition result, respectively. To visualize the output states, blue and red LEDs integrated into the polyimide flexible circuit board (PI FCB) are used to indicate $Q_{Sum}$ and $Q_{Cout}$, respectively, as shown in Supplementary Fig. 20. Figure 3f summarizes the output voltage values for all input combinations, with experimental validation provided in Supplementary Movie 8. The detailed operational mechanism of the adder is illustrated in Supplementary Fig. 21. To further validate the scalability of the system for more complex logic computing tasks (Supplementary Note 4), we fabricated a two-bit adder consisting of four inputs ($A_1$, $B_1$, $A_2$, and $B_2$)

and three outputs ($Q_{cout}$, $Q_{S2}$, and $Q_{S1}$). The experimental setup, truth table, and detailed operation process are shown in Supplementary Fig. 22. The measured results for all sixteen input combinations matched the theoretical truth table precisely, confirming the correctness, stability, and expandability of the light-programmable mechanical computing architecture.

Finally, to enable autonomous operation of the constructed computing circuits, we implemented an autonomously controlled resetting mechanism that facilitates self-regulated system recovery (Supplementary Note 5). As illustrated in Supplementary Fig. 23, when both optical inputs were off, an integrated humidity-control module was automatically activated to reset the relay, restoring its initial state for subsequent computation. Once the optical inputs resumed, the module was deactivated, allowing normal operation to continue, with experimental validation provided in Supplementary Movie 9. This autonomously controlled strategy ensures autonomous recovery without external intervention, thereby advancing the practical realization of self-sustaining, light-programmable mechanical computing systems.

### Environment-interactive camouflage enabled by light-programmable mechanical computing

Building on the demonstrated light-programmable logic computing capability, we developed an adaptive texture camouflage system inspired by cephalopods, such as octopuses and cuttlefish, which actively modulate their skin textures to blend into natural surroundings like corals, rocks, and sand. Analogously, our artificial system was designed to sense environmental optical cues and dynamically reconstruct texture patterns, offering potential applications in autonomous underwater vehicles, reconnaissance robots, and other intelligent systems requiring real-time environmental adaptability.

We constructed a $3 \times 3$ sensing-computation-emission (SCE) unit to realize texture perception and reconstruction on a flexible substrate, as shown in Fig. 4a. In the SCE unit, three single-pole single-throw (SPST) relays detect optical signals ($P_1$, $P_2$, and $P_3$) from a $3 \times 3$ region of the input image (Fig. 4a-i) and transmit the sensed information to three AND logic circuits constructed from SPST relays (Fig. 4a-ii). The circuit generates a total of nine electrical outputs ($Q_{11}$, $Q_{12}$, ..., $Q_{33}$), consisting of two signals from each AND logic circuit and three direct outputs from the SPST relay sensors. These signals drive a $3 \times 3$ light-emitting array, where the on/off states of the bulbs reproduce the luminance distribution pattern of the input image (Fig. 4a-iii). Detailed circuit configuration and operation principles are provided in Supplementary Note 6. Experimental testing confirmed the real-time sensing-computation-emission capability of the SCE unit for image perception and reconstruction (Supplementary Fig. 24 and Supplementary Movie 10).

Based on the demonstrated characteristics of the $3 \times 3$ SCE units, we evaluated the system performance within the field of view containing nine SCE units to demonstrate its texture reconstruction capability. We tested four representative brightness-gradient patterns, including gradients from bottom-left to top-right, from left to right, from bottom-right to top-left, and from bottom to top, and obtained the results shown in Fig. 4b. The reconstructed camouflage patterns (Fig. 4b-v-viii) accurately replicated the input gradients (Fig. 4b-i-iv) with high structural fidelity, validating the precise optical-to-mechanical formation of texture patterns. We further experimentally evaluated the adaptability of the system under complex natural environmental conditions. To emulate real-world camouflage conditions, we simulated three representative backgrounds inspired by coral (Fig. 4c-i), rock (Fig. 4c-ii), and sand (Fig. 4c-iii), which are typical living scenes of cephalopods. The reconstructed images (Fig. 4c-iv-vi) successfully reproduced the corresponding environmental textures, demonstrating that the light-programmable mechanical computing architecture can dynamically generate camouflage patterns resembling diverse natural surfaces.

We further analyzed the robustness of the system when exposed to environmental disturbances, such as variations in temperature, humidity, and mechanical stress. These factors may induce local failure in individual units, potentially leading to errors in the camouflage output. To assess this effect, environmental perturbations were modeled as random damage to a given proportion of functional units, which consequently lose their emissive capability. Due to the distributed design of the SCE framework, such local failures do not propagate across the system, thereby preserving global functionality. To quantitatively evaluate this robustness, we simulated camouflage performance under increasing levels of unit damage and analyzed image fidelity using a comprehensive texture similarity metric that integrates spatial and frequency-domain descriptors (Supplementary Fig. 25). The analysis reveals that even when a substantial fraction of units becomes nonfunctional, the camouflage images maintain high structural similarity to the target textures. This result highlights the intrinsic fault tolerance and resilience of the proposed design. In addition, we conducted a detailed analysis of the fidelity of the reconstructed camouflage patterns. We note that this fidelity is primarily governed by the relationship between the texture-feature size and the spatial extent of an individual SCE unit's sensing window. High-fidelity reconstruction is achieved when the texture features are substantially larger than the $3 \times 3$ sensing area, whereas features approaching this scale result in reduced reconstruction accuracy. A more detailed discussion is provided in Supplementary Note 7 (Supplementary Figs. 26–28).

Beyond functional demonstration, this study highlights a conceptual advance in environment-interactive intelligence enabled by light-programmable mechanical computing. Compared with conventional metamaterial-based mechanical computing, our approach leverages material-level programmability and thin-film integration to achieve higher switching speed, scalability, and multi-stimulus adaptability (see Supplementary Note 8, Tables 2 and 3). Moreover, the relay-based framework can be seamlessly integrated with electronic systems as an intelligent sensing-computation module. Unlike conventional sensors that merely relay external signals, our distributed mechanical network intrinsically integrates sensing and computation, providing adaptive functionality while maintaining compatibility with digital logic architectures. Collectively, these findings open a promising pathway toward hybrid intelligent materials capable of autonomous environmental perception and reconfigurable information processing.

## Discussion

In summary, we proposed a light-programmable mechanical computing scheme based on a multifunctional polyaniline composite film (PCF) that integrates diverse responsive materials into a single platform. The composite film exhibits optically induced mechanical deformation, enabling dynamic modulation of electrical pathways for light-controlled information processing. Building upon this mechanism, we constructed a scalable computing architecture composed of light-programmable relays that perform optical switching and routing, thereby realizing both fundamental and combinational logic operations, including one-bit and two-bit full adders. Furthermore, the system demonstrated environmental interactivity by sensing texture features of natural backgrounds, such as rock, coral, and sand, and generating corresponding camouflage patterns that emulate cephalopod adaptive behavior. These results collectively indicate that the proposed light-programmable mechanical computing system provides a promising paradigm for intelligent mechanical architectures capable of integrating perception and computation, and offers a feasible pathway toward mechanical computing technologies that can be practically implemented in real-world scenarios.

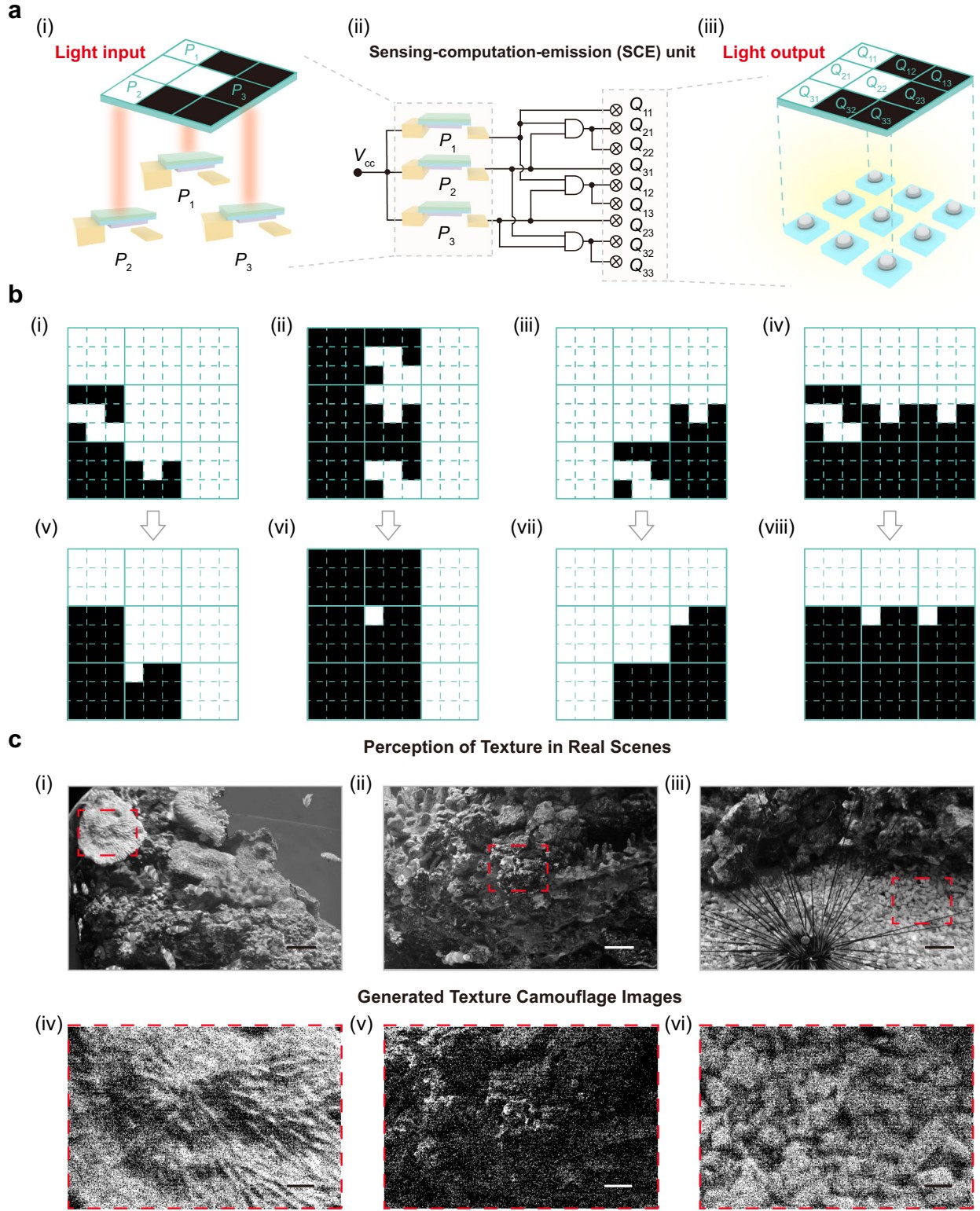

**Fig. 4 | The construction of adaptive texture camouflage. a** Process of texture camouflage: the optical information is sensed by single-pole single-throw (SPST) relays (i) within the sensing-computation-emission (SCE) unit (ii) and subsequently processed through an AND logic operation to generate control signals that drive a $3 \times 3$ light bulb array (iii). ($P_1$, $P_2$, and $P_3$ are the optical input signals, and the corresponding optical output signals are ($Q_{11}$, $Q_{12}$, ..., $Q_{33}$). **b** Evaluation of texture reconstruction: original input images (i–iv) and corresponding camouflage patterns (v–viii) generated by the computing circuit, demonstrating accurate replication of brightness gradients. **c** Realistic scenario demonstrating adaptive texture camouflage in underwater environments with corals (i), rocks (ii), and sand (iii), and simulated camouflage patterns (iv–vi) generated by the SCE unit. Scale bar: 3000 pixels (iii–iii); Scale bar: 600 pixels (iv–vi). MATLAB was used to perform simulations of the adaptive texture camouflage results.

## Methods

### Synthesis of PANi-PNIPAm

The synthesis of PANi-PNIPAm suspension follows the authors' previously published articles[58]. Synthesis of PNIPAm hydrogel: PNIPAm hydrogel was synthesized via precipitation polymerization. Solution A was prepared by dissolving N-isopropylacrylamide (NIPAm, 2.83 g, 98%, Energy Chemical), N, N′-methylene bisacrylamide (MBAm, 0.283 g, 98%, Energy Chemical), PEG-1000 (0.85 mg, Energy Chemical), and PEG-6000 (0.55 mg, Energy Chemical) in 50 mL ultrapure water, followed by $N_2$ purging (30 min). Solution B (ammonium persulfate (APS, 98%, Energy Chemical) 0.2 g in 50 mL ultrapure water, $N_2$-purged 10 min) was added to solution A. The combined solution was maintained at 70 °C for 3 h under continuous $N_2$ atmosphere with constant stirring. The polymerization was quenched by rapid cooling in an ice-water bath. The resulting milky-white PNIPAm hydrogel underwent hydration-induced expansion upon immersion. The swollen hydrogel was then homogenized at 1500 rpm for 1 h to yield a uniform PNIPAm suspension. Synthesis of PANi-PNIPAm:10 mL of hydrochloric acid, 0.93 mL of aniline (98%, TCI), 2.287 g of lithium chloride (99.5%, Energy Chemical), and 2.282 mg of APS (dissolved in 10 mL of ultrapure water) were introduced into the PNIPAm suspension, followed by a 36-hour reaction at −27 °C. The product was neutralized with $Na_2CO_3$ (99.8%, Sinopharm Chemical Reagent Co., Ltd.), washed, concentrated, and lyophilized to yield PANi-PNIPAm powder.

### Synthesis of Ag NWs

The synthesis of Ag NWs was performed by the polyol method according to the literature[63]. Briefly, 0.2 g of polyvinylpyrrolidone (PVP, $M_w$ = 360000, Energy Chemical) was added to 25 mL ethylene glycol (EG, 99.9%, Energy Chemical) under magnetic stirring. Then 0.25 g of $AgNO_3$ (99%, Sigma Aldrich) was added in the mixture to form a homogeneous solution. Finally, 3.5 g of $FeCl_3$ (98%, Energy Chemical) solution was mixed with the above solution. The resulting solution was heated to 130 °C for 5 h to form Ag NWs and then cooled to room temperature. The synthesized Ag NWs solution was purified with acetone, methanol, and ethanol, and subsequently redispersed in ethanol for further use.

### Preparation of Ag NWs/PANi-PNIPAm/PDMS

PANi-PNIPAm solution: 0.15 g of PANi-PNIPAm powder and 0.03 g of PEG-1000 were dissolved in 1.5 mL anhydrous N-methyl-2-pyrrolidone (NMP) under 6 h stirring to obtain a homogeneous dark-blue solution. PDMS film: the PDMS base and curing agent (Sylgard 184, Dow Corning) were mixed at a 10:1 weight ratio and thoroughly blended. After degassing for 40 min to remove bubbles, the mixture was coated onto glass slides and cured at room temperature for 36 h to obtain strip-shaped PDMS films. PDMS/Ag NWs: Ag NWs suspension was drop-coated onto the PDMS and thermally annealed at 80 °C. Multiple deposition cycles were performed to obtain the target Ag NWs thickness. PDMS/Ag NWs/PANi-PNIPAm: PANi-PNIPAm dispersion was dropped onto the PDMS/Ag NWs layer and thermally cured at 60 °C for 12 h to obtain a uniform PDMS/Ag NWs/PANi-PNIPAm film.

### Preparation of Ag NWs/PDMS/Ag NWs/PANi-PNIPAm

The obtained PDMS/Ag NWs/PANi-PNIPAm composite was flipped, and the Ag NWs dispersion was drop-coated onto the PDMS surface. Thermal treatment at 80 °C for 30 min produced the final Ag NWs/ PDMS/Ag NWs/PANi-PNIPAm multilayer structure.

### Fabrication of logic gates

The fabrication process for AND, OR, and XOR gates involves five key steps: (i) negative mold fabrication: negative molds for AND, OR, and XOR gates were designed using FreeCAD and fabricated via 3D printing (0.01 mm/layer resolution) with photopolymer resin (creality 3D), followed by UV-cured (1 h). (ii) SEBS substrate preparation: SEBS substrates were prepared by dissolving 5 g of styrene-ethylene-butylene-styrene (SEBS, Tuftec H1221) in 28 g of toluene (99.5%, Sinopharm Chemical Reagent Co., Ltd.) under mechanical stirring. The homogeneous solution was then cast onto negative molds (AND/OR/XOR gate) and dried in a transparent acrylic box for 4 days. Freestanding SEBS films were subsequently peeled off from the negative molds. (iii) Gold electrode deposition: Au electrodes (60–90 nm thickness) were patterned using electron-beam evaporation ($9.8 \times 10^{-5}$ Pa, 0.5 Å/s) through custom steel mesh shadow masks. (iv) Bulb integration: bulbs were attached to electrodes using silver paste and double-coated adhesive tape. (v) Polyaniline composite films (PCFs) assembly: the PCF (PDMS/Ag NWs/PANi-PNIPAm) was assembled by bonding the Ag NWs side to the electrodes with silver paste.

### Fabrication of self-adhesive SEBS connections

A 15 wt% SEBS solution (Tuftec H1221, 15 wt% in toluene) was poured into a $10 \times 10$ cm² glass slide, and then placed in a transparent acrylic box with a lid for slow drying. Gold electrodes were patterned via electron-beam evaporation (60–90 nm, 0.5 Å/s) using a laser-cut stainless-steel mesh as a shadow mask. The obtained self-adhesive SEBS connections were stored in a clean box and later segmented with a precision blade.

### The source of the image data of Fig. 4c

Figure 4c-i–iii show photographs taken by the authors and processed using the built-in *dither* function in MATLAB. This function converts grayscale images into binary (black-and-white) images while preserving perceived grayscale contrast through a dithering algorithm.

## Data availability

Source data are available on Figshare https://doi.org/10.6084/m9.figshare.30751616. All data are available from the corresponding authors upon request.

## Code availability

The Supplementary Codes used for simulation and data plotting are provided with this paper.

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

## Acknowledgements

Funding: this work was supported by the National Natural Science Foundation of China (62304105 (X.N.Y), the Fundamental Research Funds for the Central Universities KG202508 (F.M.), the National Natural Science Foundation of China (62204119 (C.P), 62304104 (Y.K.Y.), 62375130 (D.Y.), 62404099 (P.F.W.)), the National Key R and D Program of China under grant 2023YFF1203600 (S.-J.L.), the Leading-edge Technology Program of Jiangsu Natural Science Foundation (BK20232004 (F.M.)), the Natural Science Foundation of Jiangsu Province (BK20233001) (F.M.), the AI & AI for Science Project of Nanjing University (14380240 (S.-J.L.), 14380242 (F.M.)), Nanjing University International Collaboration Initiative, Open Fund of State Key Laboratory of Infrared Physics (grant no. SITP-SKLIP-ZD-2025-01 (S.-J.L.)) and Fundamental and Interdisciplinary Disciplines Breakthrough Plan of the Ministry of Education of China (JYB2025XDXM120 (F.M.)), F.M., S.-J.L. would like to acknowledge support from the AIQ foundation and the e-Science Center of Collaborative Innovation Center of Advanced Microstructures, F.M. would like to acknowledge support from the New Cornerstone Science Foundation through the XPLORER PRIZE. The microfabrication center of the National Laboratory of Solid State Microstructures (NLSSM) is also acknowledged for their technical support.

## Author contributions

F.M., S.-J.L., and C.P. conceived the idea and supervised the whole project. X.N.Y., Y.X.L., and Y.C.Z. synthesized materials, fabricated relays, and performed electrical measurements. X.N.Y., S.N.Y., conducted finite element simulations. C.P. designed the adaptive texture camouflage circuitry and performed simulations. S.N.Y., D.Y., G.-J.R., H.Z., F.Q.C., and X.-J.Y. provided assistance in the experiment. P.F.W., W.T.Y., Y.K.Y., C.W., and B.C. contributed valuable discussions. X.N.Y., C.P., S.-J.L., and F.M. co-wrote the manuscript with input from all authors.

## Competing interests

The authors declare no competing interests.

## Additional information

**Supplementary information** The online version contains Supplementary material available at https://doi.org/10.1038/s41467-026-70425-z.

