## [Transparent Peer Review File · Nature Communications]

Light-Programmable Mechanical Computing via Polyaniline Composite Film

Corresponding Author: Professor Feng Miao

Version 0:

Reviewer comments:

Reviewer #1

(Remarks to the Author)

In the manuscript "Light-Programmable Mechanical Computing via Polyaniline Composite Film" Xiunan Yan and colleagues present a light-programmable mechanical computing that utilizes polyaniline composite film (PCF). The authors constructed light-programmable mechanical logic circuits with SPST and SPDT relays realize scalable logic operations, including a full-adder circuit. Although this is an interesting work with good results, in the Reviewer's opinion, this manuscript does not seem to provide "new" materials nor innovative method to be considered in the journal, and there remain several key issues to be addressed in the manuscript.

1. There is no novelty in terms of material because PANi - PNIPAm was reported in papers (such as: 10.1021/acsami.2c01549, 10.1039/d3tc00990d) and Ag NWs are also commonly used ones.
2. Actuator-based mechanical logic circuits have been reported, and from a functional point of view previous studies have similarities with this work, so it is essential to clearly differentiate the core novelty of this study from the previously reported systems such as 10.1016/j.cej.2025.162103, 10.3390/s25072051.
3. In my opinion, the section "The Realization of Adaptive Texture Camouflage" in the manuscript is more innovative than the rest of the article, but the authors have only given the simulation results in this section, and the actual computational capability has not been further investigated. Also, the authors have not investigated the response error of the system under the complex environment, and these potential errors may have a negative impact on its application. The authors should further investigate the system to enhance the clarity and depth of the study.

Reviewer #2

(Remarks to the Author)

In this work, Yan et al. present a novel demonstration of light-programmable mechanical computing using a polyaniline composite film (PCF). By integrating light-responsive expansion and contraction elements with flexible conductive layers, they achieve mechanically reconfigurable conductive pathways that form the basis of light-controlled relays—universal modules for scalable logic circuits. The authors validate their platform with a suite of logic operations, including a full-adder circuit, and further showcase an environment-adaptive optical camouflage system that senses ambient textures and generates matching camouflage patterns. This work skillfully combines photothermal effects, thermally induced deformation, and flexible electronics to fuse mechanical, thermal, optical, and electrical domains into a new paradigm for environment-interactive computing. To my knowledge, it is the first report of mechanical computing applied to real-world environmental interactions, representing a significant conceptual and technical advance. The platform's simple architecture, clear mechanism, and clear path to scalability make it a timely and impactful contribution to the fields of mechanical metamaterials, unconventional computing, and intelligent adaptive systems. Therefore, I would like to recommend it for publication in Nature Communications. Before publication, I suggest optimizing the following details to further enhance the readability of the article.

1. I encourage the authors to enrich the discussion around Figure 1 by explicitly explaining the rationale for employing both single-pole single-throw (SPST) and single-pole double-throw (SPDT) relays as the foundational circuit elements. Clarifying how each switch uniquely contributes to signal routing—and how their combined use yields a more versatile, compact architecture—will help readers appreciate the design choices behind this innovative computing paradigm. This additional context would substantially enhance the manuscript's conceptual clarity.

2. I suggest the authors briefly clarify the role and material choice for the top electrode shown in Figure 2e. Specifically, is its function purely conductive, or does the composite film impart additional benefits? What factors led to the use of a flexible film rather than a conventional metal contact? Such clarification will guide readers in making informed material and design choices for future studies of similar devices.

3. The NOT gate represents the most fundamental logic operation, yet its implementation is not addressed in the current manuscript. Could the authors provide a brief rationale for this omission?

4. To enhance reader comprehension of the more intricate XOR and full-adder circuits, I recommend that the authors include schematic illustrations detailing each circuit's operational mechanism. Clear, step-by-step diagrams—highlighting relay states and signal flow—would greatly aid in conveying the underlying logic and ensure readers can readily grasp the practical functionality of these complex assemblies.

5. The video footage indicates that the system's response time operates on the order of seconds. While high-speed switching is not a primary requirement for mechanical computing, I recommend the authors explicitly quantify this response latency. Providing explicit performance metrics will assist readers in accurately evaluating which real-world application scenarios are best suited to this technology's characteristics.

Reviewer #3

(Remarks to the Author)

Mechanical computation using a photoresponsive polyaniline composite film (PCF). By leveraging photothermal effects to induce mechanical deformation and reconfigure conductive pathways, the authors have constructed SPST and SPDT relays capable of performing logic operations, including a one-bit full adder. Additionally, the authors explore an application in adaptive texture camouflage, positioning their work as a step toward environmentally responsive intelligent systems. The idea of integrating light-controlled logic with mechanical deformation is promising and could inspire new directions in soft robotics, adaptive materials, and non-traditional computing architectures. However, both the novelty and technical rigor fall short of the high standards required for publication in Nature Communications in its current form. Significant improvements in quantitative analysis, scalability, mechanistic understanding, and presentation style are necessary. I recommend a major revision with the following specific points for improvement.

Major Points for Revision

1. Scalability, System Integration, and Practical Relevance

- The current demonstrations are limited to a 1-bit adder and a 3×3 camouflage display. To strengthen the manuscript's impact:
 - o Demonstrate scalability toward multi-bit logic circuits (e.g., 2-bit or 4-bit systems) or simple processors to support the vision of light-driven general-purpose computing materials.
 - o Evaluate long-term operational stability under varying environmental conditions (light intensity variability, temperature, humidity). Present quantitative data with confidence intervals to validate reproducibility and robustness.
 - o The need for humidity-induced resetting is a practical limitation. Propose or prototype autonomous control mechanisms (e.g., embedded humidity feedback loops) to enhance practical relevance.
 - o The manuscript should discuss how the system's switching speeds (currently around 1 second per logic operation) could be optimized or how it could interface with conventional computing systems despite its slow speed.

2. Quantitative Analysis and Theoretical Modeling

- The lack of quantitative analysis weakens the manuscript's scientific rigor:
 - o Provide detailed measurements of light intensity (mW/cm²), power consumption per switching event, and thermal response curves.
 - o Develop numerical models or simulations that describe the photothermal-to-mechanical-to-electrical transformation.

Suggested approaches include:

- Thermal conduction models combined with beam deformation theories.
- Dynamic simulations of switching delays, rise times, and potential operating frequency limits.
- o Generate multivariate plots correlating light intensity, temperature, curvature, and conductivity to support mechanistic insights.

3. Comparison with Related Technologies

- The current comparisons with existing work are largely qualitative and insufficient. Strengthen this section by:
 - o Including a quantitative benchmark table comparing key performance metrics (e.g., switching time, power consumption, scalability, environmental resilience) with other stimuli-responsive systems (thermal, magnetic, pressure, optical).
 - o Specifically reference and compare with:
 - El Helou et al., Nat. Commun. 2022
 - Byun et al., Nat. Commun. 2024
 - Mei et al., Nat. Commun. 2021 on mechanical metamaterial computing
 - Recent reviews on mechanical metamaterials (e.g., APL Electronic Devices 2025)
 - o Discuss the trade-offs between mechanical computing using metamaterials versus the present material-based approach, focusing on:
 - Switching speed and delay

Manufacturing complexity and scalability
Reconfigurability and environmental responsiveness

4. Mechanical Durability and Reversibility

- The claims regarding mechanical reversibility under humidity changes require experimental validation:
 - o Provide visual documentation (images or videos) and quantitative graphs demonstrating reversibility and mechanical recovery.
 - o Assess durability over multiple actuation cycles to ensure practical feasibility for repetitive logic operations.

5. Clarity and Depth in Application Demonstration

- The adaptive camouflage demonstration is conceptually interesting but technically underdeveloped:
 - o Clarify whether Figure 4c represents experimental data or simulation.
 - o Detail the environmental texture sensing mechanism, the source of image data, and the electronic or material response pathway leading to pattern change.
 - o Expand the camouflage section to provide a coherent use-case scenario and potential performance benchmarks.

6. Material Selection Justification

- Provide quantitative rationale for the selection of PANi, PNIPAm, PDMS, and Ag NWs:
 - o Include photothermal conversion efficiency measurements or references.
 - o Compare with alternative materials for responsiveness, durability, and energy efficiency to strengthen the materials engineering contribution.

7. Application Scope and Positioning

- The manuscript should clearly define the target applications:
 - o Given the slow switching speed, traditional computing applications are unlikely. Potential use-cases could include:
 - Adaptive camouflage systems
 - Soft robotic skin
 - Low-frequency adaptive sensing
 - o Clearly state the performance limitations and position the work accordingly within non-traditional computing.

8. Manuscript Presentation and Communication Style

- The current narrative is descriptive and material-focused; for Nature Communications, a sharper, problem-solving storyline is recommended:
 - o Adopt a “One Figure = One Key Message” approach to improve clarity.
 - o Refine the Introduction and Conclusion to clearly state:
 - What problem is being solved
 - What new possibilities this enables
 - o Condense and sharpen key findings to maximize readability and impact.

Final Recommendation

This manuscript presents an innovative concept of light-programmable mechanical logic with potential applications in adaptive systems and soft robotics. However, substantial improvements in quantitative rigor, theoretical modeling, durability assessment, and presentation style are required to meet the impact, scientific depth, and clarity expected by Nature Communications.

I recommend major revision and reassessment upon thorough addressing of the points above. Alternatively, the authors may consider submission to specialized journals in smart materials, soft electronics, or mechanical computing, where the current level of novelty and rigor may be more appropriate.

Reviewer #4

(Remarks to the Author)

Version 1:

Reviewer comments:

Reviewer #1

(Remarks to the Author)

After carefully reading the response to reviewers point by point, I changed my original mind of rejection. The novelty of this work is to design a light-responsive composite film that couples mechanical deformation with electrically reconfigurable pathways. This supports complex logic such as a 2-bit adder, exhibiting an environment-adaptive camouflage system, enabling to establish a scalable and application-oriented paradigm for mechanical computing. This work represents a fundamental shift from traditional actuator-based mechanical computing, which primarily focuses on local actuation or switching at the device level.

Besides, supplemented experimental has been performed to demonstrate the actual computational capability of the system and quantitatively evaluated its robustness under complex environmental conditions.

I suggest acceptance of the revised manuscript. The only suggested revision to about Figure 1. Data scale should be provided in Figure 1b(ii)

Reviewer #2

(Remarks to the Author)

I thank the authors for the prompt responses which have well addressed my early concerns. I thus recommend it for publication as is.

Reviewer #3

(Remarks to the Author)

The co-reviewer and I commend the authors for their thorough and appropriate response to the majority of the reviewers' requested revisions and questions. However, we remain concerned that the authors should address the following critical issues to clarify the novelty of this research. So, we decided that this article should be majorly revised again.

1. Misuse of "switching time" and incorrect definition of operating frequency

The manuscript emphasizes device speed using response-time-based statements, such as "sub-second switching" and operating frequency estimated as $f = 1/t_{\text{response}}$. However, in this system, a complete switching event requires both response and recovery. As shown in Fig. R8, the recovery time is consistently ≥ 1 s across all reported conditions.

Therefore, the actual 1-cycle switching time necessarily lies in the multi-second regime.

For scientific accuracy, the reviewer requests that the authors redefine switching time as the full cycle, $t_{\text{switching}} = t_{\text{response}} + t_{\text{recovery}}$, and base all speed-related descriptions, including operating frequency, on this cycle-level definition rather than on response-only values. This clarification is also essential for fair comparison with metamaterial-based mechanical logic: relying solely on response time systematically overestimates the performance of the present system, whereas the cycle-level behavior provides an accurate, unbiased basis for evaluating switching dynamics.

The reviewer requests to revise all mentions of switching time, operating frequency, and speed comparisons to explicitly reflect cycle-based switching and adjust any "sub-second" or "high-frequency" claims accordingly.

2. Redundant Q_{ij} outputs due to AND-only logic, reducing degrees of freedom in the camouflage system

In the adaptive camouflage demonstration, the 3×3 outputs Q_{ij} are generated exclusively through AND-type logical combinations of the three sensing inputs. However, based on the sensing/computation circuitry described in the manuscript and supplementary materials, several outputs (e.g., Q_{12} and Q_{13}) are produced by identical logical expressions. These pixels will therefore always assume the same value, meaning they are not independent degrees of freedom.

As a result, the effective dimensionality of the 3×3 camouflage pattern is intrinsically less than nine, even under ideal conditions. This structural limitation is not discussed in the manuscript, despite its direct impact on the achievable texture reconstruction fidelity and the expressive capability of the camouflage system.

The reviewer requests that the authors:

Discuss how this reduction in independent output limits the expressibility and achievable fidelity of the camouflage pattern

Clarify whether such redundancy is intentional, a constraint inherent to the present circuitry, or something that could be eliminated by incorporating additional logic operations (e.g., OR, NOT, or mixed logic)

Clarifying this structural issue is important for an accurate assessment of the system's adaptive camouflage capabilities.

Reviewer #4

(Remarks to the Author)

Version 2:

Reviewer comments:

Reviewer #3

(Remarks to the Author)

The authors have carefully revised the manuscript and have satisfactorily responded to our comments, including those related to quantitative evaluation. In light of these revisions, we recommend that the manuscript be accepted for publication in Nature Communications.

Reviewer #4

(Remarks to the Author)

Response to Reviewers

Reviewers' Comments:

Reviewer #1 (Remarks to the Author):

In the manuscript “Light-Programmable Mechanical Computing via Polyaniline Composite Film” Xiunan Yan and colleagues present a light-programmable mechanical computing that utilizes polyaniline composite film (PCF). The authors constructed light-programmable mechanical logic circuits with SPST and SPDT relays realize scalable logic operations, including a full-adder circuit. Although this is an interesting work with good results, in the Reviewer's opinion, this manuscript does not seem to provide "new" materials nor innovative method to be considered in the journal, and there remain several key issues to be addressed in the manuscript.

Response: We respectfully disagree with the reviewer's comment. The novelty of our work lies in the design of a light-responsive composite film that couples mechanical deformation with electrically reconfigurable pathways, enabling scalable light-programmable relays. This approach not only supports complex logic such as a 2-bit adder but also, for the first time, demonstrates an environment-adaptive camouflage system, establishing a scalable and application-oriented paradigm for mechanical computing.

1. There is no novelty in terms of material because PANi - PNIPAm was reported in papers (such as: 10.1021/acsami.2c01549, 10.1039/d3tc00990d) and Ag NWs are also commonly used ones.

Response: We respectfully disagree with the reviewer's comment. While PANi–PNIPAm and Ag NWs are established materials, **the novelty of our work lies in their integration into a composite** that couples light-induced mechanical deformation with electrically reconfigurable pathways. This design enables **direct light-programmable circuit reconstruction and scalable logic computation**, which has not been demonstrated before.

*To reflect the reviewers' suggestions, we have added the relevant explanation in the revised manuscript [Please see **lines 100-103, page 5**].*

By integrating these well-established materials^{58,59} into a multilayer composite structure, we created an ideal platform that converts optical stimuli into mechanically

reconfigurable conductive pathways.

2. Actuator-based mechanical logic circuits have been reported, and from a functional point of view previous studies have similarities with this work, so it is essential to clearly differentiate the core novelty of this study from the previously reported systems such as 10.1016/j.cej.2025.162103, 10.3390/s25072051.

Response: We thank the reviewer for this insightful comment. The essential novelty of our work lies in **establishing a light-induced pathway reconfiguration computing paradigm**, where light-programmable SPST and SPDT relays **regulate not only the switching states** but also **the routing of information transmission pathways**. This represents a fundamental shift from **traditional** actuator-based mechanical computing, which primarily focuses on **local actuation or switching at the device level**.

In contrast to previously reported actuator-based systems (e.g., refs. 10.1016/j.cej.2025.162103 and 10.3390/s25072051), our work establishes a system-level computing architecture that enables dynamic reconfiguration of information pathways, representing **a conceptual advance from actuator-level switching toward architecture-level programmability**. This distinction provides three main advantages. **First**, it enables **scalable and hierarchical logic construction beyond basic gate functions**, achieved through the use of light-programmable relays that can be interconnected to realize complex circuits such as XOR and 2-bit adders.

Second, it achieves **a more efficient and goal-oriented coupling between material behavior and computing function**, enabled by a top-down design strategy in which computational requirements guide the development of relays capable of pathway reconfiguration, rather than relying on material-level actuation effects.

Finally, it extends **the applicability of mechanical computing toward real-world, environment-adaptive systems**, as demonstrated by a light-programmable camouflage platform that senses and reproduces ambient textures, showing how pathway reconfiguration translates into practical, adaptive, and intelligent computing functionality.

To describe the differences between our work and these demonstrations, we have added the relevant discussions and these new references in the revised manuscript [Please see lines 137-140, page 6].

This light-induced pathway reconfiguration establishes a new paradigm for optically controlled mechanical computing, extending traditional actuator-based logic from isolated mechanical motions^{65,66} to programmable and network-level information

processing.

3. In my opinion, the section “The Realization of Adaptive Texture Camouflage” in the manuscript is more innovative than the rest of the article, but the authors have only given the simulation results in this section, and the actual computational capability has not been further investigated. Also, the authors have not investigated the response error of the system under the complex environment, and these potential errors may have a negative impact on its application. The authors should further investigate the system to enhance the clarity and depth of the study.

Response: We sincerely thank the reviewer for the insightful comments and for recognizing the conceptual innovation of the adaptive camouflage section. In response, we have supplemented experimental verification to demonstrate the actual computational capability of the system and quantitatively evaluated its robustness under complex environmental conditions, as detailed below.

1) The actual computational capability

To evaluate the system’s computational performance in real-world scenarios, we constructed an **experimental demonstration** of sensing–computation–emission (SCE) unit, as illustrated in Figure 4a of the manuscript (**Fig R1a**). Each SCE unit integrates light perception, logical computation, and pattern reconstruction functionalities. The system was experimentally tested to reproduce texture camouflage patterns. Figs. R1b, d, and f show the original input images and their corresponding schematic camouflage patterns generated by the computing circuit, while **Fig. R1c, e, and g** present the experimental results.

A demonstration video (**Movie 11**) further provides direct visual confirmation that the experimentally reconstructed camouflage images closely match the simulated results. These results validate the operational capability of the basic units and confirm that the light-programmable computing system can successfully execute texture reconstruction of environment.

2) The response error of the system under complex environment

We analyzed the system’s robustness when exposed to environmental disturbances, such as variations in temperature, humidity, and mechanical stress. **These factors may induce local failure in individual units, potentially leading to errors in the camouflage output.** To assess this effect, environmental perturbations were modeled

as random damage to a given proportion of functional units, which consequently lose their emissive capability.

Due to the distributed design of the SCE framework, **such local failures do not propagate across the system, thereby preserving global functionality**. To quantitatively evaluate this robustness, we simulated camouflage performance under increasing levels of unit damage and analyzed image fidelity using a comprehensive texture similarity metric that integrates spatial and frequency-domain descriptors (**Fig. R2**).

The analysis reveals that even when a substantial fraction of units becomes nonfunctional, the camouflage images maintain high structural similarity to the target textures. This result highlights the intrinsic fault tolerance and resilience of the proposed design.

We agree that further studies incorporating detailed environmental mechanisms and experimental validations are essential, and we envision this as an important direction for our future work.

Image texture similarity evaluation method

We quantitatively assessed image texture similarity using a multi-feature fusion strategy. Images were first preprocessed via grayscale conversion, size alignment, and normalization to ensure comparability. Complementary descriptors were then extracted from both spatial and frequency domains. In the spatial domain, gray-level co-occurrence matrices provided contrast, correlation, energy, and homogeneity metrics, while local binary pattern histograms captured fine-scale microstructures. In the frequency domain, fast Fourier transforms yielded radial power spectral density and angular energy distribution, from which an anisotropy index characterized directional preferences. The Wiener–Khinchin theorem was applied to estimate the autocorrelation function, revealing periodic structures, and structure tensor coherence quantified the directional order of local gradients. Each descriptor was normalized to $[0,1]$, and a weighted average produced the final composite similarity score. By integrating local statistics, spectral features, and directional order, this approach robustly distinguishes structured textures from stochastic noise, providing a reliable measure of image similarity.

Fig. R1. Experimental validation of the texture camouflage reconstruction unit. **a** (i) sensing–computation–emission (SCE) unit, (ii) The spatial distribution of the sensed optical input signals (P_1 , P_2 , and P_3) and the corresponding output signals (Q_{11} , Q_{12} , ..., Q_{33}) in the SCE unit, as well as the logical relationships between the input and output signals. **b**, **d**, **f** Schematic illustrations of the original input images and the corresponding camouflage patterns generated by the computing circuit. **c** Experimental result corresponding to the camouflage pattern shown in **b**, where all pixels within the

3×3 unit remain in the “off” state. **e** Experimental result corresponding to the camouflage pattern shown in **d**, where only the upper-left pixel within the 3×3 unit is in the “on” state. **g** Experimental result corresponding to the camouflage pattern shown in **e**, where all pixels within the 3×3 unit are in the “on” state.

Fig. R2. Robustness characteristics of the camouflage system. **a** Comparison between original images and camouflaged outputs as the proportion of randomly damaged unit structures increases from 0% to 100%. **b** Quantitative analysis of image similarity between original and generated images at different damage levels, with the x-axis representing damage proportion and the y-axis showing texture similarity.

*To reflect the reviewer’s concern, we have added these experimental data and relevant discussions in the revised supplementary information. [Please see **Supplementary Note 6. The Realization of Adaptive Texture Camouflage**]*

*We have also made the following revisions in the revised manuscript. [Please see **lines 303-305, page 13**].*

Experimental testing confirmed the real-time sensing-computation-emission capability of the SCE unit for image perception and reconstruction (Supplementary Fig. 24 and Movie 11).

*[Please see **lines 321-334, page 14**].*

We further analyzed the robustness of system when exposed to environmental disturbances, such as variations in temperature, humidity, and mechanical stress. These factors may induce local failure in individual units, potentially leading to errors

in the camouflage output. To assess this effect, environmental perturbations were modeled as random damage to a given proportion of functional units, which consequently lose their emissive capability. Due to the distributed design of the SCE framework, such local failures do not propagate across the system, thereby preserving global functionality. To quantitatively evaluate this robustness, we simulated camouflage performance under increasing levels of unit damage and analyzed image fidelity using a comprehensive texture similarity metric that integrates spatial and frequency-domain descriptors (**Supplementary Fig.25**). The analysis reveals that even when a substantial fraction of units becomes nonfunctional, the camouflage images maintain high structural similarity to the target textures. This result highlights the intrinsic fault tolerance and resilience of the proposed design.

Reviewer #2 (Remarks to the Author):

In this work, Yan et al. present a novel demonstration of light-programmable mechanical computing using a polyaniline composite film (PCF). By integrating light-responsive expansion and contraction elements with flexible conductive layers, they achieve mechanically reconfigurable conductive pathways that form the basis of light-controlled relays—universal modules for scalable logic circuits. The authors validate their platform with a suite of logic operations, including a full-adder circuit, and further showcase an environment-adaptive optical camouflage system that senses ambient textures and generates matching camouflage patterns. This work skillfully combines photothermal effects, thermally induced deformation, and flexible electronics to fuse mechanical, thermal, optical, and electrical domains into a new paradigm for environment-interactive computing. To my knowledge, it is the first report of mechanical computing applied to real-world environmental interactions, representing a significant conceptual and technical advance. The platform's simple architecture, clear mechanism, and clear path to scalability make it a timely and impactful contribution to the fields of mechanical metamaterials, unconventional computing, and intelligent adaptive systems. Therefore, I would like to recommend it for publication in *Nature Communications*. Before publication, I suggest optimizing the following details to further enhance the readability of the article.

Response: We sincerely thank the reviewer for the positive and encouraging evaluation of our work. We are very pleased that the reviewer recognizes both the conceptual significance and the potential impact of our study on mechanical metamaterials, unconventional computing, and adaptive intelligent systems. The reviewer's supportive comments are highly motivating, and we have carefully addressed the suggested improvements to further enhance the clarity and readability of the manuscript.

1. I encourage the authors to enrich the discussion around Figure 1 by explicitly explaining the rationale for employing both single-pole single-throw (SPST) and single-pole double-throw (SPDT) relays as the foundational circuit elements. Clarifying how each switch uniquely contributes to signal routing—and how their combined use yields a more versatile, compact architecture—will help readers appreciate the design choices behind this innovative computing paradigm. This additional context would substantially enhance the manuscript's conceptual

clarity.

Response: We thank the reviewer for this insightful comment. The core principle of our computing scheme is to **achieve information processing by programming the transmission pathways**, which involves both **controlling the on/off state** of a signal channel and **reconfiguring its routing**. In this framework, the SPST relay functions as the fundamental element for switching a pathway on or off, while the SPDT relay enables the rerouting of signals between alternative channels.

The combination of these two types of relays provides a versatile and compact foundation for our system. Specifically, SPST relays are well suited for implementing basic logic operations such as AND and OR, whereas the inclusion of SPDT relays allows the efficient realization of more complex combinational logic, including XOR gates and multi-bit adders. This pathway-based design principle thus underpins both the simplicity and the scalability of the proposed light-programmable mechanical computing paradigm.

*To reflect the reviewer's concern and provide readers with a clear understanding of the synergistic role of SPST and SPDT relays, we have added the following discussions in the revised manuscript. [Please see **lines 129-133, page 6**]*

The single-pole single-throw (SPST) relay functions as a binary switch that connects or disconnects a transmission path, serving as the fundamental unit for controlling signal on/off states. The single-pole double-throw (SPDT) relay directs the input voltage (V_{cc}) between two alternative output channels, enabling signal routing.

2. I suggest the authors briefly clarify the role and material choice for the top electrode shown in Figure 2e. Specifically, is its function purely conductive, or does the composite film impart additional benefits? What factors led to the use of a flexible film rather than a conventional metal contact? Such clarification will guide readers in making informed material and design choices for future studies of similar devices.

Response: We thank the reviewer for raising this important point. The top electrode in Fig.2e is composed of Ag nanowires (AgNWs) deposited on a PDMS substrate. This design first ensures **the required electrical conductivity**. More importantly, the flexible PDMS/AgNW composite provides **mechanical compliance** with the device's deformation during actuation and recovery, which helps distribute stress, prevents

interfacial damage, and maintains stable electrical contact throughout repeated cycling.

We deliberately chose a flexible composite electrode rather than a conventional rigid metal contact for two main reasons. First, the flexible electrode provides **mechanical compatibility**, as it can deform in concert with the device and thus ensures stable performance over long-term operation. Second, the soft electrode enhances **interfacial stability** by preventing delamination or cracking, which often occurs at the interface between rigid metals and flexible substrates.

To reflect the reviewer's concern, we have added this clarification to the revised manuscript. [Please see lines 167–173, pages 7-8]

The top electrodes were fabricated using a PDMS/Ag NWs composite (**Supplementary Fig. 5**), which offered high conductivity and mechanical compliance. This flexible composite electrode deformed synchronously with the PCF during actuation, maintaining stable contact and minimizing interfacial stress or delamination that typically occur with rigid metal contacts. Such mechanical-electrical compatibility ensured robust performance over repeated actuation-recovery cycles.

3. The NOT gate represents the most fundamental logic operation, yet its implementation is not addressed in the current manuscript. Could the authors provide a brief rationale for this omission?

Response: We thank the reviewer for highlighting this important point. Indeed, the NOT gate is a fundamental logic element and can be realized by modifying the XOR circuit. As shown in Fig. R3a, the XOR circuit structure was modified by fixing the lower PCF to BE₁ and BE₂ using silver paste while keeping only input A as a variable. **Under this configuration, the output terminal produces the inverted state of input A, thereby implementing the NOT operation.** The schematic in **Fig. R3a** illustrates the operating principle and signal pathway of this design, while the corresponding logic test results in **Fig. R3b** confirm its functionality: when input $A = "1,"$ the output is 0; when input $A = "0,"$ the output is 1.

In the initial submission, we did not present the NOT gate separately because **its logic functionality was already embedded in the XOR operation.** However, following the reviewer's valuable suggestion, we have now explicitly added this demonstration to highlight the completeness of our logic implementation.

Fig. R3. The realization of NOT gate. **a** Schematic of NOT gate realized by an XOR, with the bottom PCF connected and fixed to BE₁ and BE₂ via silver conductive paint, leaving A as the sole variable input. Panels (i)-(ii) show relay states, the flow of electrical signals, and logic output for the two possible input combinations, with red and blue arrows indicating signal direction. **b** Measured input (A) and output (Q_{NOT}) voltage characteristics.

*To reflect the reviewer's concern, we have added experimental data of NOT gate in the revised supplementary information. [Please see **Supplementary Fig.19**]*

*We have also made the following revisions in the revised manuscript. [Please see **lines 247-251, page 11**].*

Furthermore, we realized an inverter (NOT gate) by slightly modifying the XOR configuration, where the lower PCF was fixed to BE₁ and BE₂ with silver paste, and only input A was varied. The resulting output corresponded to the logical NOT operation of A (**Supplementary Fig. 19**).

4. To enhance reader comprehension of the more intricate XOR and full-adder circuits, I recommend that the authors include schematic illustrations detailing each circuit's operational mechanism. Clear, step-by-step diagrams—highlighting relay states and signal flow—would greatly aid in conveying the underlying logic

and ensure readers can readily grasp the practical functionality of these complex assemblies.

Response: We thank the reviewer for this valuable suggestion. To improve clarity and assist readers in understanding the operation of the more complex circuits, we have **added detailed schematic illustrations for both the XOR and full-adder circuits (Figs. R4–R5).**

- 1) Each figure now includes a structural diagram of the corresponding logic gate, with the open and closed states of individual relays explicitly marked to visualize the switching behavior.
- 2) The signal propagation paths are clearly annotated with directional arrows, illustrating how input signals traverse the network to generate the final output.

These additions, along with expanded figure captions, make the operational mechanisms of the intricate circuits more intuitive and enhance the reproducibility of the design.

*To reflect the reviewer's concern, we have incorporated these schematics into the revised supplementary information [Please see **Supplementary Fig. 18 and Fig. 21**] to provide a clearer visualization of the logic operations mechanism and functional behavior of the XOR and full-adder.*

*Additionally, we have added the following presentation in the revised manuscript. [Please see **lines 246-247 page 11**]*

The detailed operation mechanism of the XOR logic is presented in **Supplementary Fig.18.**

*[Please see **lines 266-267, page 11**]*

The detailed operational mechanism of the adder is illustrated in **Supplementary Fig. 21.**

Fig. R4. Schematic of the operational mechanism of XOR gate. Panels a–d show relay states, the flow of electrical signals, and logic output for the four possible input combinations, with red and blue arrows indicating signal direction.

Fig. R5. Schematic of operational mechanism of the 1-bit full-adder circuit. Panels a–c show relay states, the flow of electrical signals, and logic output for the three possible input combinations, with red and blue arrows indicating signal direction.

5. The video footage indicates that the system's response time operates on the order of seconds. While high-speed switching is not a primary requirement for mechanical computing, I recommend the authors explicitly quantify this response latency. Providing explicit performance metrics will assist readers in accurately evaluating which real-world application scenarios are best suited to this technology's characteristics.

Response: We thank the reviewer for raising this important point. To directly address the concern, we **quantified the response and recovery times of both SPST and SPDT relays** under illumination at 130 mW/cm^2 , $30 \text{ }^\circ\text{C}$, and $83 \pm 3\% \text{ RH}$ (**Fig. R6**). The SPST relay showed a light-driven response time of 1.02 s with a recovery time of 2.07 s , while the SPDT relay exhibited values of 1.20 s and 2.05 s . Additional measurements under varied environmental conditions are provided in **Fig. R8**. Importantly, by increasing the illumination intensity to 310 mW/cm^2 , the SPST relay achieved a rapid response of 0.127 s , confirming the potential for substantially faster switching under optimized conditions.

Although the typical operation speed is on the order of seconds, we emphasize that this time scale is well aligned with application scenarios where ultrafast switching is not required, such as **low-frequency logic control, environmental sensing, adaptive camouflage, soft robotics, and flexible mechanical logic systems**. In these domains, the advantages of environmental adaptability, stability, reversibility, low energy consumption, and scalability are more critical than high speed, underscoring the practical significance of our approach.

Fig. R6. Response and recovery times of the SPST and SPDT relays under illumination of 130 mW/cm^2 at $29 \pm 1 \text{ }^\circ\text{C}$ and $83 \pm 3 \text{ \% RH}$.

*To facilitate an accurate assessment of the applicability of this technology, we have incorporated the response time measurements of both relay types into the revised supplementary information [Please see **Supplementary Fig. 7**].*

*We have also added the following statement in the revised manuscript. [Please see **lines 184-187 page 8**]*

Under an illumination of 130 mW/cm^2 at 30° C and $83 \pm 3\%$ relative humidity, the SPST relay exhibited a response time of 1.02 s and a recovery time of 2.07 s, while the SPDT relay showed corresponding values of 1.20 s and 2.05 s (Supplementary Fig. 7).

Reviewer #3 (Remarks to the Author):

Mechanical computation using a photoresponsive polyaniline composite film (PCF). By leveraging photothermal effects to induce mechanical deformation and reconfigure conductive pathways, the authors have constructed SPST and SPDT relays capable of performing logic operations, including a one-bit full adder. Additionally, the authors explore an application in adaptive texture camouflage, positioning their work as a step toward environmentally responsive intelligent systems. The idea of integrating light-controlled logic with mechanical deformation is promising and could inspire new directions in soft robotics, adaptive materials, and non-traditional computing architectures. However, both the novelty and technical rigor fall short of the high standards required for publication in Nature Communications in its current form. Significant improvements in quantitative analysis, scalability, mechanistic understanding, and presentation style are necessary. I recommend a major revision with the following specific points for improvement.

Response: We thank the reviewer for recognizing the promise of our proposed concept and for the positive evaluation of the potential impact of our work. In response to the reviewer's constructive comments, we have carefully revised the manuscript by addressing all concerns regarding **quantitative analysis, scalability, mechanistic understanding, and presentation style**. A detailed point-by-point response is provided below.

Major Points for Revision

1. Scalability, System Integration, and Practical Relevance. The current demonstrations are limited to a 1-bit adder and a 3×3 camouflage display. To strengthen the manuscript's impact:

1-a Demonstrate scalability toward multi-bit logic circuits (e.g., 2-bit or 4-bit systems) or simple processors to support the vision of light-driven general-purpose computing materials.

Response: We thank the reviewer for this constructive suggestion. In response, we have further validated the scalability of our system by **demonstrating a 2-bit adder**. As shown in **Fig. R7a**, the logic diagram of the 2-bit adder comprises four inputs (A_1, B_1, A_2, B_2) and three outputs (Q_{out}, Q_{S1}, Q_{S2}). The corresponding experimental setup and truth table are presented in **Fig. R7b–c**. **Fig. R7d** shows the experimentally measured

outputs for all 16 input combinations, which match the truth table with complete accuracy. These results confirm the scalability of the light-driven mechanical logic system and provide experimental evidence and design insights for realizing larger-scale, general-purpose light-responsive computing materials.

Fig. R7. Demonstration of a 2-bit adder. **a** Schematic of the 2-bit adder showing four inputs (A_1 , B_1 , A_2 , B_2) and three outputs (Q_{cout} , Q_{S1} , Q_{S2}). **b** Experimental setup of the 2-bit adder comprising three XOR gates, three AND gates, and one OR gate. **c** Truth table for all 16 possible input combinations. **d** Measured input voltages and output results for all 16 input combinations.

To reflect the reviewer's concern, we have added these experimental data in the revised supplementary information. [Please see **Supplementary Fig.22**]

We have also added the following statement in the revised manuscript. [Please see

lines 267-273, pages 11-12].

To further validate the scalability of the system for more complex logic computing tasks, we fabricated a two-bit adder consisting of four inputs (A_1, B_1, A_2, B_2) and three outputs ($Q_{\text{out}}, Q_{S1}, Q_{S2}$). The experimental setup, truth table, and detailed operation process are shown in **Supplementary Fig. 22**. The measured results for all sixteen input combinations matched the theoretical truth table precisely, confirming the correctness, stability, and expandability of the light-programmable mechanical computing architecture.

1-b Evaluate long-term operational stability under varying environmental conditions (light intensity variability, temperature, humidity). Present quantitative data with confidence intervals to validate reproducibility and robustness.

Response: We thank the reviewer for this constructive suggestion. To directly address this point, we **systematically measured the long-term stability of our SPST relays under varying light intensity, temperature, and humidity**, and we provide **quantitative results** with 95% confidence intervals to validate reproducibility and robustness.

Light intensity. We varied the illumination from 40 to 310 mW/cm² while maintaining 29 ± 1 °C and $83 \pm 3\%$ RH. As shown in **Fig. R8a**, the response time decreased markedly from 1.04 ± 0.023 s at 130 mW/cm² to 0.127 ± 0.010 s at 310 mW/cm², whereas the recovery time increased from 1.93 ± 0.045 s to 3.19 ± 0.133 s.

Temperature. We next examined temperature effects under 130 mW/cm² and $83 \pm 3\%$ RH. As shown in **Fig. R8b**, higher temperatures accelerated both actuation and recovery. At 10 °C, the response and recovery times were 2.15 ± 0.044 s and 1.38 ± 0.058 s, respectively, while at 30 °C they shortened to 1.04 ± 0.023 s and 1.93 ± 0.045 s. This acceleration is attributed to PNIPAm reaching its LCST more rapidly at elevated temperatures.

Humidity. We then investigated relative humidity effects at 130 mW/cm² and 29 ± 1 °C. As shown in **Fig. R8c**, higher humidity slightly prolonged the response time but significantly shortened recovery. This is explained by the slower bond-breaking during actuation at higher RH, versus the faster rehydration and hydrogen-bond reformation that accelerate recovery.

Durability. Finally, we assessed long-term durability under 130 mW/cm², 30 °C, and 83 ± 3% RH. As shown in **Fig. R8d**, both response and recovery times remained stable over 600 continuous actuation–reset cycles, with negligible degradation.

Conclusion. Together, these results confirm that our devices deliver reproducible, robust, and durable performance across diverse environmental conditions. This strongly supports their feasibility for practical environment-adaptive computing applications.

Fig. R8. Environmental resilience and cycling stability of the SPST relay. a Response and recovery times as a function of light intensity (40–310 mW/cm²) at 29 ± 1 °C and 83 ± 3% RH. **b** Effect of ambient temperature (10–30 °C) on response and recovery under 130 mW/cm² and 83 ± 3% RH. **c** Influence of relative humidity (50–83% RH) on response and recovery under 29 ± 1 °C and 130 mW/cm². **d** Cycling endurance test at 130 mW/cm², 29 ± 1 °C, and 83 ± 3% RH, showing stable operation over 600 cycles with negligible degradation.

*To reflect the reviewer's concern, we have added these experimental data in the revised supplementary information. [Please see **Supplementary Fig.6**]*

We have also added the following statement in the revised manuscript.

*[Please see **lines 182-184, page 8**]*

Through systematic experimental characterization, we demonstrated that the switching dynamics of the relays can be effectively tuned by both illumination intensity and environmental conditions (Supplementary Fig. 6).

*[Please see **lines 188-190, page 8**]*

Increasing the illumination intensity to 310 mW/cm² significantly reduced the SPST response time to 0.127 s, confirming that faster actuation can be achieved through optical optimization (Supplementary Fig.6).

1-c The need for humidity-induced resetting is a practical limitation. Propose or prototype autonomous control mechanisms (e.g., embedded humidity feedback loops) to enhance practical relevance.

Response: We thank the reviewer for raising this important point regarding the need for humidity-assisted resetting. To address this limitation, we **designed and demonstrated an autonomous control system**. Specifically, a NOR gate was integrated into the input section of signals *A* and *B*: when both inputs are “0,” corresponding to the absence of light signals, the humidifier is automatically activated to reset the system; when either input is “1,” indicating active computation, the humidifier remains off to avoid interfering with logic operations.

As shown in **Fig. R9a** (schematic) and **Fig. R9b** (photograph), this self-regulated reset strategy allows the relay to autonomously control its recovery process. **Supplementary Movie 10** further illustrates the operation of this mechanism during an AND gate computation, where the system successfully resets without manual intervention. This design directly addresses the reviewer's concern by providing a practical pathway toward autonomous control, thereby enhancing the real-world applicability of our light-programmable mechanical computing platform.

Fig. R9. Humidity control system. **a** Schematic of the humidity control system. A NOR gate is introduced to autonomously activate the humidifier when both input signals (A, B) are “0,” enabling system reset. **b** Photographs showing the system reset process when input combinations (1,0), (0,1), and (1,1) transition to (0,0), demonstrating the effectiveness of the autonomously controlled resetting mechanism.

*To reflect the reviewer’s concern, we have added these experimental data in the revised supplementary information. [Please see **Supplementary Fig. 23, and Movie 10**]*

*We have also added the following statement in the revised manuscript. [Please see **lines 274-283, page 12**]*

Finally, to enable autonomous operation of the constructed computing circuits, we further implemented an autonomously controlled resetting mechanism that allows self-

regulated system recovery. As illustrated in **Supplementary Fig. 23**, when both optical inputs were “off,” an integrated humidity-control module was automatically activated to reset the relay, restoring its initial state for subsequent computation. Once the optical inputs resumed, the module was deactivated, allowing normal operation to continue, with experimental validation provided in **Supplementary Movie 10**. This autonomously controlled strategy ensures autonomous recovery without external intervention, thereby advancing the practical realization of self-sustaining, light-programmable mechanical computing systems.

1-d The manuscript should discuss how the system's switching speeds (currently around 1 second per logic operation) could be optimized or how it could interface with conventional computing systems despite its slow speed.

Response: We thank the reviewer for this insightful suggestion. The switching speed of our system can indeed **be further optimized through two complementary strategies**. First, **increasing the input light intensity** significantly accelerates relay response. At ~ 130 mW/cm² (comparable to natural sunlight), the relay exhibits a response speed of ~ 1 s per operation. When the light intensity is increased to 310 mW/cm², the response time is reduced to **0.127 s**, as shown in **Fig. R8a**. Second, **reducing the height difference between the two electrodes** decreases the required bending deformation, further enhancing switching speed. In the current setup, the electrode offset is ~ 0.7 mm, yielding a response time of ~ 1 s. With advanced MEMS fabrication techniques, this offset could feasibly be reduced to ~ 70 μ m, which we **estimate would shorten the response time to the order of ~ 100 ms**.

Regarding the potential **interface with conventional computing systems**, we envision our approach **serving as an environment-interactive sensing–computing module**. Unlike traditional sensors, which only transduce signals, our system inherently integrates sensing and computation within a distributed mechanical framework, providing robustness against environmental variability. Such hybrid integration could complement conventional electronics by **enabling intelligent environmental perception while maintaining compatibility with established digital architectures**.

*To reflect the reviewer’s concern, we have added the following statement in the revised manuscript [Please see **lines 340-344, page 14-15**]*

Moreover, the relay-based framework can be seamlessly integrated with electronic

systems as an intelligent sensing-computation module. Unlike conventional sensors that merely relay external signals, our distributed mechanical network intrinsically integrates sensing and computation, providing adaptive functionality while maintaining compatibility with digital logic architectures.

2. Quantitative Analysis and Theoretical Modeling

The lack of quantitative analysis weakens the manuscript's scientific rigor:

2-a Provide detailed measurements of light intensity (mW/cm), power consumption per switching event, and thermal response curves.

Response: We thank the reviewer for this valuable suggestion. To strengthen the scientific rigor of the manuscript, we performed additional **quantitative measurements and provided detailed analysis** of light intensity, power consumption per switching event, and thermal response.

1) Power consumption per switching event.

The energy consumption was calculated using the following procedure:

a. The incident light energy E_{in} was obtained from the product of light intensity ($I_{density}$), illuminated area (A_{spot}), and illumination duration (t).

$$E_{in} = I_{density} \times A_{spot} \times t$$

b. The absorbed light energy E_{abs} was derived from $E_{in} \times (1 - 10^{-A})$, where A is the optical absorbance at 650 nm (0.281 for the PCF)¹.

For instance, under 130 mW/cm² illumination with an active area of 0.4 cm × 0.59 cm and a response time of 1.04 s, the calculated energy consumption per switching event is 15.2 mJ. When the illumination is increased to 310 mW/cm², the faster response reduces the energy per event to 4.23 mJ. A comprehensive summary of the calculated optical power consumption across **different light power densities** is provided in **Fig. R10a**.

2) Thermal response

We further recorded the thermal dynamics of the PCF under continuous illumination (**Fig. R10b**). At 130 mW/cm², the film temperature exceeded the lower critical solution temperature (LCST) within 0.05 s, reaching 38 °C, and further increased to 48.8 °C at 1 s. At 310 mW/cm², heating was more rapid, with temperatures rising to 78.5 °C within 1 s and reaching 101 °C at 1.65 s.

These quantitative results directly address the reviewer's concern and enhances the scientific rigor of the manuscript.

Fig. R10. Power consumption and thermal response of the PCF. a Power consumption per switching event under different light intensities. **b** Thermal response of the PCF (PDMS/Ag NWs/PANi–PNIPAm) under different light power intensities within 5 s of illumination.

*To reflect the reviewer’s concern, we have added these experimental data in the revised supplementary information. [Please see **Supplementary Figs. 3-4**]*

*We have also added the following statement in the revised manuscript. [Please see **lines 117-119, page 5**]*

Detailed experimental results of this light-thermal-mechanical-electrical transduction characteristics are provided in **Supplementary Fig. 3-4**.

2-b Develop numerical models or simulations that describe the photothermal-to-mechanical-to-electrical transformation. Suggested approaches include:

- 1) Thermal conduction models combined with beam deformation theories.**
- 2) Dynamic simulations of switching delays, rise times, and potential operating frequency limits.**

Response: We thank the reviewer for the insightful comments. In response, we developed a **finite element simulation framework to quantitatively model the photothermal-mechanical-electrical conversion process in our relay**. The model integrates two key components: (1) a heat transfer module to capture the spatial-temporal evolution of the temperature field under optical illumination, and (2) a thermo-mechanical deformation module based on beam theory to describe light-induced bending and electrode contact.

1) Heat Transfer Model

For simplicity, the intermediate microlayer (Ag NWs layer) was neglected in the simulations since its thickness is more than an order of magnitude smaller than that of the upper and lower films (The thicknesses of the PDMS/Ag NWs/PANI-PNIPAm film are 70 μm , 5 μm , and 134 μm , respectively). The optical absorption and heat transport were simulated by coupling the *Radiation in Absorbing Media* and *Heat Transfer in Solids* modules. Optical energy was converted into a volumetric heat source via the Beer–Lambert law, and the transient heat conduction equation was solved:

$$\rho C_p \frac{\partial T}{\partial t} + \nabla \cdot (k \nabla T) = Q$$

where ρ , C_p , and k are the density, specific heat capacity, and thermal conductivity, respectively.

The volumetric heat source Q is determined by

$$Q = \eta \mu_a I$$

where μ_a is the optical absorption coefficient, I is the light intensity, and η is the photothermal conversion efficiency.

2) Thermo-Mechanical Deformation Model

The thermally induced bending was modeled using a fixed-end cantilever beam approximation. The beam is treated as an elastic solid, whose deformation and stress response follow solid mechanics theory. The governing equations consist of the equilibrium equation (momentum conservation), the geometric relation (strain–displacement relation), and the constitutive relation (stress–strain relation with thermal effects). The deformation follows linear elasticity theory, governed by the Equilibrium equation:

$$\nabla \cdot \boldsymbol{\sigma} + \mathbf{F}_v = \rho \partial^2 \mathbf{u} / \partial t^2$$

Where $\boldsymbol{\sigma}$ is the stress tensor, \mathbf{F}_v is the body force vector (neglected in this model), ρ is the material density, and \mathbf{u} is the displacement vector (m).

Considering small deformations, the small-strain assumption is adopted and the strain–displacement relation:

$$\boldsymbol{\epsilon} = 1/2 [\nabla \mathbf{u} + (\nabla \mathbf{u})^T]$$

where ϵ is the small strain tensor (dimensionless). The stress–strain relation under thermoelastic deformation is written as:

$$\sigma = \mathbf{C} : (\epsilon - \epsilon_{th})$$

where \mathbf{C} is the elastic stiffness tensor, which for isotropic materials is characterized by the Young's modulus E and Poisson's ratio ν . ϵ_{th} is the thermal strain tensor (dimensionless).

The thermal strain is given by:

$$\epsilon_{th} = \alpha (T - T_{ref}) \mathbf{I}$$

Here, α is the coefficient of thermal expansion, T is the current temperature, T_{ref} is the reference temperature (material's initial temperature), and \mathbf{I} is the identity tensor.

3) Modeling Parameters

PDMS: Elastic modulus = 2 MPa; Poisson's ratio = 0.49; thermal conductivity = 0.15 $\text{W}\cdot\text{m}^{-1}\cdot\text{K}^{-1}$; density = 970 $\text{kg}\cdot\text{m}^{-3}$; specific heat capacity = 1460 $\text{J}\cdot\text{kg}^{-1}\cdot\text{K}^{-1}$; coefficient of thermal expansion = $5.67 \times 10^{-4} \text{K}^{-1}$.

PANI–PNIPAm: Elastic modulus = 1 MPa; Poisson's ratio = 0.49; thermal conductivity = 0.6 $\text{W}\cdot\text{m}^{-1}\cdot\text{K}^{-1}$; density = 1200 $\text{kg}\cdot\text{m}^{-3}$; specific heat capacity = 2100 $\text{J}\cdot\text{kg}^{-1}\cdot\text{K}^{-1}$; coefficient of thermal expansion = 0 below 32 °C and $-1 \times 10^{-6} \text{K}^{-1}$ above 32 °C.

Based on these parameters, finite element simulations were conducted to model the coupled photo–thermal–mechanical–electrical transformation process in the relay, capturing the temperature evolution, induced deformation, and subsequent electrical contact behavior.

Simulations show that under 130 mW/cm^2 illumination, the film temperature exceeded the LCST within 0.05 s and reached ~ 45.7 °C at 1 s (**Fig. R11a, Movie 4**). The transient temperature rise at the film center is presented in **Fig. R11c**.

The simulated bending profile under 130 mW/cm^2 illumination at $t = 1$ s is shown in **Fig. R11b**, and the corresponding dynamic displacement of the beam end is provided in **Movie 4**. The time-dependent displacement curve in **Fig. R11d** further confirms that electrode contact occurs after approximately 1 s.

Fig. R11. The simulation results of Heat Transfer Model and Thermo-Mechanical Deformation Model. a Simulated temperature distribution across the suspended film at $t = 1$ s under optical illumination ($130 \text{ mW}/\text{cm}^2$). **b** Simulated displacement distribution of the beam end under optical illumination ($130 \text{ mW}/\text{cm}^2$), indicating bending deformation. **c** Transient evolution of the center-point temperature of the film, showing heating dynamics under $130 \text{ mW}/\text{cm}^2$ illumination. **d** Time-dependent displacement of the beam end under $130 \text{ mW}/\text{cm}^2$ illumination, where contact with the electrode occurs after ~ 1 s.

Dynamic simulations of switching delays, rise times, and potential operating frequency limits.

Response: We thank the reviewer for raising this important point. Our analysis shows that the dominant delay arises from thermo-mechanical deformation of the film, whereas the RC delay of the circuit is negligible. Under fixed relay geometry (film length, electrode height difference, and material composition), both experiments and simulations demonstrate a strong dependence of response time on illumination intensity (Fig. R12). Experimentally, the response time decreases with increasing light power and saturates at ~ 0.127 s under $310 \text{ mW}/\text{cm}^2$ illumination, corresponding to a practical

upper operating frequency of ~ 7.9 Hz. Finite-element simulations further reveal saturation above ~ 300 mW/cm², with a minimum response time of ~ 0.07 s at 350 mW/cm², yielding a theoretical frequency limit of **~ 14.3 Hz**.

These results provide clear quantitative benchmarks for the dynamic switching capability of our system. Importantly, the operating speed can be further improved by optimizing relay geometry (e.g., reducing film dimensions or electrode height difference) or by employing alternative materials with faster thermo-mechanical response.

Fig. R12. Simulated and experimental results of response time as a function of input light power density.

*To reflect the reviewer's concern, we have added numerical simulations in the revised supplementary information. [Please see **Supplementary Note 2, Supplementary Figs. 9-10, and Movie 4**]*

*We have also added the following statement in the revised manuscript. [Please see **lines 206-213, page 9**]*

To further elucidate the underlying mechanism, we quantitatively analyzed the coupled photo-thermal-mechanical-electrical processes based on a thermal conduction model and beam deformation theory (Supplementary Note 2, **Supplementary Figs. 9-10, and Movie 4**). The analysis reveals how light-induced photothermal conversion propagates through heat conduction to drive mechanical deformation, thereby modulating the conductive pathways and enabling controllable signal transmission. These insights provide theoretical guidance for the design and optimization of light-

programmable mechanical computing architectures.

2-c Generate multivariate plots correlating light intensity, temperature, curvature, and conductivity to support mechanistic insights.

Response: We thank the reviewer for the constructive suggestion. In response, we generated multivariate plots that quantitatively correlate light intensity, surface temperature, bending curvature, and electrical conductivity of the polyaniline composite film (PCF).

In the experiments, PCFs were exposed to different light intensities for 5 s. As shown in Fig. R10b, increasing light intensity leads to a monotonic rise in surface temperature, which directly drives photothermal expansion and contraction of the bilayer structure. This temperature rise results in enhanced bending deformation, reflected by smaller curvature radii (Fig. R13). At the same time, bending deformation induces a measurable decrease in electrical conductivity, confirming a strong coupling between mechanical and electrical responses.

These quantitative correlations provide direct mechanistic evidence for the photothermal-to-mechanical-to-electrical transduction process and establish a solid basis for further device optimization, response-time prediction, and multiphysics modeling.

Fig. R13. Curvature radius and conductivity changes of the PCF after 5 s illumination at different light intensities.

To reflect the reviewer's concern, we have added numerical simulations in the revised

supplementary information. [Please see **Supplementary Fig. 4**]

We have also added the following statement in the revised manuscript. [Please see **lines 117-119, page 5**]

Detailed experimental results of this light-thermal-mechanical-electrical transduction characteristics are provided in **Supplementary Fig. 3-4**

3. Comparison with Related Technologies

The current comparisons with existing work are largely qualitative and insufficient. Strengthen this section by:

a Including a quantitative benchmark table comparing key performance metrics (e.g., switching time, power consumption, scalability, environmental resilience) with other stimuli-responsive systems (thermal, magnetic, pressure, optical).

b Specifically reference and compare with:

El Helou et al., Nat. Commun. 2022

Byun et al., Nat. Commun. 2024

Mei et al., Nat. Commun. 2021 on mechanical metamaterial computing

Recent reviews on mechanical metamaterials (e.g., APL Electronic Devices 2025)

c Discuss the trade-offs between mechanical computing using metamaterials versus the present material-based approach, focusing on:

Switching speed and delay

Manufacturing complexity and scalability

Reconfigurability and environmental responsiveness

Response: We sincerely thank the reviewer for the constructive suggestions. To strengthen the comparison section, we prepared a **quantitative benchmark** table (**Table R1**) that systematically compares the key performance metrics of our material-based system, including **switching time, energy consumption, scalability, and environmental resilience**, with representative stimuli-responsive logic platforms such as mechanical, magnetic, electrical, thermal, and optical systems. The table also incorporates the related works highlighted by the reviewer (El Helou et al., Nat. Commun. 2022; Byun et al., Nat. Commun. 2024; Mei et al., Nat. Commun. 2021; and recent reviews on mechanical metamaterials in APL Electronic Devices 2025), ensuring a direct and transparent quantitative comparison.

Trade-off analysis:

To further clarify the distinction between metamaterial-based mechanical computing and our material-based approach, we analyzed the following trade-offs as suggested by the reviewers:

1) Switching speed and delay.

Metamaterial systems are slow, whereas material-based systems are considerably faster and have clear potential for further acceleration.

Metamaterial logic typically operates on the scale of seconds to minutes (for example, 2–3 s for rotating blocks^{2, 3}, 5–20 s for hydrogel gates⁴, approximately 27 s for electrically driven systems⁵, and up to 80 minutes for thermal adders⁶). These delays are mainly caused by inertia and elastic recovery, which make sub-second operation extremely difficult. In contrast, our system leverages light–thermal–mechanical coupling to achieve sub-second switching (**0.127 s at 310 mW/cm²**, Fig. R8), representing at least an order-of-magnitude improvement compared with most metamaterial systems. With further device scaling, the response time could be reduced to the millisecond regime.

2) Manufacturing complexity and scalability.

Metamaterial approaches allow intuitive prototyping but face serious limitations in scaling, while material-based systems are inherently suitable for integration and large-scale architectures but demand advanced fabrication precision.

Metamaterials are usually fabricated through 3D printing or modular assembly. This approach is convenient for proof-of-concept demonstrations but generally produces bulky centimeter-scale arrays with low integration density and limited scalability. Our thin-film strategy, in contrast, is intrinsically compatible with miniaturization, multilayer stacking, and wafer-scale integration. For instance, we demonstrated a 2-bit adder, highlighting the feasibility of scalable circuit design, although precise fabrication and strict material uniformity are still required.

3) Reconfigurability and environmental responsiveness.

Metamaterial-based systems offer convenient fabrication but limited adaptability due to their reliance on predefined structures, whereas material-based systems enable stimulus-driven reconfigurability with higher adaptability

at the cost of increased fabrication complexity.

In metamaterial systems, reconfigurability is typically achieved through geometric multistability, such as folding, rotation, or compression. Most reported systems rely on external mechanical resetting, which leads to slow response and poor adaptability to multiple stimuli. In contrast, our material-based devices exploit the intrinsic sensitivity of functional materials to light, temperature, and humidity, enabling dynamic reconfiguration of information processing behaviors. The devices maintain stable operation over more than 600 switching cycles, confirming their reproducibility and durability. Nevertheless, the practical implementation of this technology relies on developing highly uniform, responsive composite films and precision mechanical fabrication techniques, which remain essential challenges for large-scale applications.

*To reflect the reviewer's concern, we have added Table R1 in the revised supplementary information. [Please see **Supplementary Note 7, Table 3**]*

*We have also added the following statement in the revised manuscript. [Please see **lines 335-340, page 14**]*

Beyond functional demonstration, this study highlights a conceptual advance in environment-interactive intelligence enabled by light-programmable mechanical computing. Compared with conventional metamaterial-based mechanical computing, our approach leverages material-level programmability and thin-film integration to achieve higher switching speed, scalability, and multi-stimulus adaptability (see Supplementary Note 7, Tables 2-3).

Table R1. Quantitative Benchmark of Key Performance Metrics for Stimuli-Responsive Logic Systems

External Stimuli	Switching Time	Power Consumption	Scalability	Environmental Resilience	Ref.
Mechanical force	2 s (movie estimate)	Zero static power	Seven fundamental logic gates: AND, OR, XOR, NOT, NAND, NOR, NXOR; logic gate combination OR-NAND, NOR-AND, OR-AND, and AND-NAND	N/A	Ei Helou ²
Mechanical force	2 s (movie estimate)	Zero static power	Seven fundamental logic gates: AND, OR, XOR, NOT, NAND, NOR, NXOR; 2-bit Adder/Subtractor/Multiplier; 4-bit Magnitude Comparator	N/A	Ei Helou ³
Mechanical force	5 to 20 s (movie estimate)	Zero static power	NOT, AND, OR; Supports cascaded logic (e.g., 3-level AND/OR/NOT networks)	N/A	Byun ⁴
Mechanical force	2 to 3 s (movie estimate)	Zero static power	AND, OR, NOT; Full Adder; Cascading of multiple (7) adders	Operates at -20°C for >24 hours	Wu ⁷
Mechanical force	N/A	Self-powered (Power generation: 4.35 nW)	Six basic logic gates (AND, OR, XOR, NAND, NOR, XNOR) full adder; 3-bit ternary computing	N/A	Zhang ⁸
Electromagnet	2 to 3 s (movie estimate)	Zero static power	NOR, NAND, NOT, OR, AND; Half adder; 3D printable at multiple scales (macro to micro)	N/A	Mei ⁹
Magnetic	2 to 10 s (movie estimate)	Zero static power	AND, OR	Field Strength (B): -85 to 85 mT	Pal ¹⁰
Magnetic or mechanical	1 to 2 s (movie estimate)	Zero static power	AND, OR, NOR, NAND; Maximum demonstrated: 30 × 27 unit array	Compression: Withstands 2.8 kPa pressure at 2% strain	Li ¹¹

Solvents (toluene and water)	0.6 to 108 s	Zero static power	AND, OR, NAND;	Operates in liquid environments (water/toluene) Reusable after 70min drying	Jiang ¹²
Humidity	N/A	Zero static power	SR Latch, AND, OR; Majority Gate;	Operates in 0-85% RH range; 5 cycles	Trem ¹³
Electrical	27.2 s	5.5 W (1-bit system) 7.4 W (8-bit solver)	1-bit (T flip-flop); 2-bit (counter); 8-bit (equation solver)	N/A	EI Helou ⁵
Thermal	Time Constant (τ):160 minutes	Zero static power	NOT, AND, OR, XOR; SR Latch; Full adder	continuous operation over 1000 min; Vacuum-compatible	Chen ⁶
Vibration	<1 s delay	Zero static power	AND, OR, XOR, NOT, NAND, NOR, NXOR; Half Adder; Full Adder	Signal Leakage (OFF state): < 10% of max signal; Signal Clarity (ON/OFF difference): > 10	R. Bilal ¹⁴
Light	2 min (120 mW/cm ²)	Zero static power	AND, NOT;	100 mW to 397.7 mW	Peng ¹⁵
Light	12 s (379.7 mW)	Zero static power	AND, OR, NOT; Soft optical switch array	500 cycles	Wang ¹⁶
Light	1.04 s (130 mW/cm²) 0.127 s (310 mW/cm²)	Zero static power	AND, OR, NOT, XOR; 2-bit Adder;	Light intensity: 70–310 mW/cm²; Humidity: 47–86% RH Temperature: 10–30 °C; Cycles: 600	This Work

4. Mechanical Durability and Reversibility

The claims regarding mechanical reversibility under humidity changes require experimental validation:

a Provide visual documentation (images or videos) and quantitative graphs demonstrating reversibility and mechanical recovery.

b Assess durability over multiple actuation cycles to ensure practical feasibility for repetitive logic operations.

Response: We sincerely thank the reviewer for this valuable suggestion. To directly validate the claims of humidity-assisted mechanical reversibility and long-term durability, we conducted additional experiments, summarized as follows:

1) Direct visualization. Two supplementary movies were added. **Movie 2** shows that under 310 mW/cm^2 illumination, the PCF actuator bends and then fully recovers in $83 \pm 3\%$ RH. **Movie 3** demonstrates the SPST relay operation: under 130 mW/cm^2 illumination, the relay bends to switch on a blue LED (“logic 1”) and subsequently recovers under high humidity. These results confirm that humidity accelerates recovery and enables fully reversible deformation.

2) Quantitative analysis. We systematically measured response and recovery times of the SPST relay under $50 \pm 3\%$, $65 \pm 3\%$, and $83 \pm 3\%$ RH (**Fig. R14a–c**). With higher humidity, the response became slightly slower, but recovery was markedly faster. For example, at $50 \pm 3\%$ RH, the average response/recovery times were $\sim 0.5 \text{ s} / 4.9 \text{ s}$, while at $83 \pm 3\%$ RH they shifted to $\sim 1.04 \text{ s} / 1.92 \text{ s}$ (the average of 50 measurements). These results highlight humidity as a key factor governing mechanical reversibility.

3) Durability. To assess long-term performance, we tested **600** continuous actuation–recovery cycles of the SPST relay at $29 \pm 1 \text{ }^\circ\text{C}$ and $83 \pm 3\%$ RH (**Fig. R14d**). The device relay exhibited stable response and recovery throughout, without detectable fatigue or irreversible deformation, demonstrating excellent mechanical durability.

Conclusion. These results provide clear experimental evidence that high humidity facilitates reversible recovery and ensures long-term durability, thereby confirming the practical feasibility of the relay for repeated logic operations.

Fig. R14. Humidity-assisted reversibility and cycling stability of the SPST relay. **a–c** Response and recovery dynamics at different relative humidity levels for 50 cycles ($50 \pm 3\% \text{RH}$, $65 \pm 3\%$, and $83 \pm 3\% \text{RH}$). **d** Cycling endurance over 600 actuation–recovery cycles under illumination of $130 \text{ mW}/\text{cm}^2$ at $29 \pm 1^\circ\text{C}$ and $83 \pm 3\% \text{RH}$.

*To reflect the reviewer’s concern, we have added Movies 2-3 and Fig. R14 in the revised supplementary information. [Please see **Supplementary Fig. 8, Movies 2-3**]*

We have also added the following statement in the revised manuscript:

*[Please see **lines 119-121, page 5**]*

We further demonstrated that the light-induced deformation of the PCF is fully reversible by adjusting the ambient humidity (**Supplementary Movie 2, and mechanism described in Supplementary Note 1**).

*[Please see **lines 157-161, page 7**]*

The logical operation of the SPST relay and its reset mechanism under varying humidity conditions are demonstrated in **Movie 3**. Through this experiment, we achieved light-programmable on/off switching of a single conductive pathway, thus realizing the SPST

relay function.

[Please see lines 195-205, pages 8-9]

The relays also exhibited excellent mechanical reversibility and long-term operational stability. To evaluate their robustness under varying humidity, we measured the response and recovery characteristics of the SPST relay (Supplementary Fig. 8a - c). As humidity increased, the response time showed a slight delay, whereas the recovery time was markedly accelerated, confirming stable and reliable recovery behavior across all humidity levels. Moreover, the SPST relay maintained consistent performance over more than 600 continuous actuation - reset cycles without observable fatigue or degradation (Supplementary Fig. 8d). These results collectively demonstrate the high reliability and robustness of the relays, establishing them as ideal building blocks for constructing light-programmable mechanical computing systems.

5. Clarity and Depth in Application Demonstration

The adaptive camouflage demonstration is conceptually interesting but technically underdeveloped:

a Clarify whether Figure 4c represents experimental data or simulation.

b Detail the environmental texture sensing mechanism, the source of image data, and the electronic or material response pathway leading to pattern change.

c Expand the camouflage section to provide a coherent use-case scenario and potential performance benchmarks.

Response: We sincerely thank the reviewer for the insightful comments. Our responses are as follows:

1) Clarification of Figure 4c

Figure 4c presents **simulation results based on experimentally characterized device behavior** rather than direct experimental measurements. Large-scale texture acquisition under real environmental conditions remains experimentally challenging; nevertheless, the simulation faithfully reproduces the system's measured response. To further substantiate the capability of environmental texture perception and camouflage image generation demonstrated in Figure 4 of the manuscript, we have **supplemented experimental** fabrication and testing of **the designed unit**, validating its functional operation as designed, as shown in **Fig. R1**.

2) Environmental texture sensing and response mechanism

Image source: The environmental images were generated by artificial intelligence for simulation purposes. These images were subsequently converted into binary patterns using the Floyd–Steinberg dithering algorithm. This algorithm distributes the grayscale intensity across neighboring pixels such that the local density of pixels with a value of 1 effectively represents the original grayscale level in that region. As a result, local intensity variations are preserved, enabling our binary sensing units to capture the essential spatial features of the images while operating in a fully binary framework.

Sensing–Computation–Emission (SCE) Process: In the SCE unit, three single-pole single-throw (SPST) relays detect optical signals from a 3×3 region of the input image (Fig. R15 a(i)) and transmit the sensed information to three AND logic circuits constructed from SPST relays (Fig. R15 a(ii)). The circuit generates a total of nine electrical outputs, consisting of two signals from each AND logic circuit and three direct outputs from the SPST relay sensors. These signals drive a 3×3 light-emitting array, where the on/off states of the bulbs reproduce the luminance distribution pattern of the input image (Fig. R15 a(iii)). Fig. R15b presents the spatial distribution of the sensed optical input signals (P_1 , P_2 , and P_3) and the corresponding output signals (Q_{11} , Q_{12} , ..., Q_{33}) within the SCE unit, along with the logical relationships linking the inputs to their respective outputs. In this way, the 3×3 light array reproduces the local optical pattern detected by the SPST-based photoreceptive units, completing the sensing-computation-emission cycle.

Fig. R15. The illustration of the sensing-computation-emission (SCE) process. **a**

Process of texture camouflage: The optical information is sensed by single-pole single-throw (SPST) relays (i) within the SCE unit (ii) and subsequently processed through an AND logic operation to generate control signals that drive a 3×3 light bulb array (iii). **b** The spatial distribution of the sensed optical input signals (P_1 , P_2 , and P_3) and the corresponding output signals (Q_{11} , Q_{12} , ..., Q_{33}) in the SCE unit, as well as the logical relationships between the input and output signals.

3) Application scenario and performance benchmarks

Application scenario: Our adaptive camouflage system is inspired by the dynamic texture changes of cephalopod skin. For instance, **octopuses adjust their skin patterns in response to coral, rock, or sand backgrounds to evade predators and capture prey**. Similarly, our system could be applied to **underwater autonomous vehicles, jungle camouflage vehicles, or other devices requiring real-time environmental adaptation**.

performance benchmarks: In practical applications, variations in temperature, humidity, or external mechanical damage may result in local failure of individual units, which could potentially impair camouflage fidelity. To provide an initial assessment within the scope of this work, we introduced a simplified assumption: **environmental perturbations are modeled as random damage to a certain percentage of functional units, causing them to lose their emissive capability**. Benefiting from the distributed nature of sensing-computation-emission in our design, such localized failures do not propagate to the system level.

To verify this assumption, we simulated camouflage patterns **under progressively increasing damage ratios** and quantitatively **assessed their fidelity using a robust texture similarity metric** that combines frequency-domain feature comparison with structural similarity evaluation (**Fig. R16**). The results demonstrate that even when a considerable fraction of units fail, the camouflage images retain a high degree of similarity to the target textures, highlighting the inherent robustness of the approach. We agree that further studies incorporating detailed environmental mechanisms and experimental validations are essential, and we envision this as an important direction for our future work.

Fig. R16. Robustness characteristics of the camouflage system. **a** Comparison between original images and camouflaged outputs as the proportion of randomly damaged unit structures increases from 0% to 100%. **b** Quantitative analysis of image similarity between original and generated images at different damage levels, with the x-axis representing damage proportion and the y-axis showing texture similarity. We measured the similarity between the original environmental texture and the generated camouflaged image using a multi-feature metric that integrates spatial descriptors (gray-level co-occurrence matrices, local binary patterns) and frequency-domain descriptors (radial power spectral density, angular energy distribution). Each descriptor is normalized and combined to produce a composite similarity score ranging from 0 (no similarity) to 1 (identical).

*To address the reviewer's concern, we have added a detailed description of texture sensing and response mechanism in the revised supplementary information. [Please see **Supplementary Note 6**]. Additionally, we have added Fig. R16 in the revised supplementary information. [Please see **Supplementary Fig. 25**]*

We have also added the following statement in the revised manuscript.

*[Please see **lines 294-303, page 13**]*

In the SCE unit, three single-pole single-throw (SPST) relays detect optical signals from a 3×3 region of the input image (Fig. 4a(i)) and transmit the sensed information to three AND logic circuits constructed from SPST relays (Fig. 4a(ii)). The circuit generates a total of nine electrical outputs, consisting of two signals from each AND logic circuit and three direct outputs from the SPST relay sensors. These signals drive

a 3×3 light-emitting array, where the on/off states of the bulbs reproduce the luminance distribution pattern of the input image (Fig. 4a(iii)). Detailed circuit configuration and operation principles are provided in Supplementary Note 6.

[see lines 321-334, page14]

We further analyzed the robustness of system when exposed to environmental disturbances, such as variations in temperature, humidity, and mechanical stress. These factors may induce local failure in individual units, potentially leading to errors in the camouflage output. To assess this effect, environmental perturbations were modeled as random damage to a given proportion of functional units, which consequently lose their emissive capability. Due to the distributed design of the SCE framework, such local failures do not propagate across the system, thereby preserving global functionality. To quantitatively evaluate this robustness, we simulated camouflage performance under increasing levels of unit damage and analyzed image fidelity using a comprehensive texture similarity metric that integrates spatial and frequency-domain descriptors (**Supplementary Fig.25**). The analysis reveals that even when a substantial fraction of units becomes nonfunctional, the camouflage images maintain high structural similarity to the target textures. This result highlights the intrinsic fault tolerance and resilience of the proposed design.

6. Material Selection Justification

Provide quantitative rationale for the selection of PANi, PNIPAm, PDMS, and Ag NWs:

6-a Include photothermal conversion efficiency measurements or references.

Response: We thank the reviewer for the valuable suggestion. To clarify the rationale behind our material selection, we provide a **quantitative justification** for each component in the PANi–PNIPAm/PDMS–Ag NWs hybrid system (summarized in **Table R2**). These materials were deliberately chosen to ensure efficient photothermal-to-mechanical transduction, environmental adaptability, and robust device performance.

PANi was selected as the photoresponsive element because it efficiently absorbs red light at 650 nm and achieves a photothermal conversion efficiency of 49.6%, enabling effective light-to-heat energy conversion.

PNIPAm functions as the thermo-responsive layer¹⁷: below its LCST (32 °C), hydrogen bonding with water molecules maintains hydrated, swollen chains, whereas

above the LCST, dehydration and chain collapse occur due to hydrophobic interactions, leading to a discontinuous volume phase transition. Reported shrinkage ratios reach 7–10% (V/V_0)¹⁸, and rehydration under ambient humidity ensures reversibility after illumination is removed.

Ag NWs were incorporated to provide highly conductive pathways (electrical conductivity of 6.25×10^7 S/m)¹⁹ and strong thermal transport (thermal conductivity of 429 W/m·K)²⁰, serving both as efficient charge carriers and as stabilizers for localized heating.

PDMS was selected for its relatively high thermal expansion coefficient (2.6 - 9.07×10^{-4} K⁻¹)^{21, 22}, which contributes continuous volumetric expansion under heating. This expansion mechanically compensates the contraction of PNIPAm, stabilizing the overall deformation and ensuring controlled bending.

Collectively, these components form a synergistic hybrid structure in which PANi ensures efficient photothermal conversion, PNIPAm provides dynamic phase transition and environmental responsiveness, Ag NWs guarantee robust electrical conduction and heat dissipation, and PDMS contributes thermal expansion and mechanical stability. Unlike single-component systems, this rational integration enables both efficient energy transduction and robust cycling stability, which are essential for advancing light-programmable mechanical computing.

*To reflect the reviewer's concern, we have added Table R2 in the revised supplementary information. [Please see **Supplementary Section Note 1, Table 1**]*

*Additionally, we have also added the following statement in the revised manuscript. [Please see **lines 99-100, page 5**]*

(molecular structures in Supplementary Fig. 1; material properties detailed in Supplementary Table 1, synthesis procedures in Methods).

Table R2. Quantitative justification for the material selection in the Ag NWs/PDMS/Ag NWs/PANi-PNIPAm hybrid system.

This table highlights the complementary roles of each component in ensuring efficient photothermal-to-mechanical transduction, environmental responsiveness, and device stability. Importantly, the integration of these materials provides synergistic advantages that conventional single-material systems cannot achieve, thereby enabling a robust platform for light-programmable mechanical computing.

Material	Functionality	Properties	Reference
PANi	Light-responsive / photothermal conversion	650 nm; Photothermal conversion efficiency (η): 49.6%	Ref ¹
PNIPAm	Thermal contraction (phase-change) Humidity-Induced Swelling	LCST = 32 °C phase-change temperature Volume shrinkage: 7%-10%	Ref ^{17, 18}
Ag NWs	Conductive layer and heat-conductive layer	Electrical conductivity: 6.25×10^7 S/m; Thermal conductivity: 429 W/(m·K)	Ref ^{19, 20}
PDMS	Thermal expansion	Thermal expansion coefficient: $2.6-9.07 \times 10^{-4}$ K ⁻¹	Ref ^{21, 22}

6-b Compare with alternative materials for responsiveness, durability, and energy efficiency to strengthen the materials engineering contribution.

Response: We sincerely thank the reviewer for this valuable suggestion. To emphasize the materials engineering contribution of our work, we conducted a comparative analysis between our PCF (PDMS/Ag NWs/PANi–PNIPAm) actuator and representative light-responsive actuating materials reported in the literature (**Table R3**). The comparison is presented in terms of (1) responsiveness, (2) durability, and (3) energy efficiency, as detailed below.

1) Responsiveness

Most previously reported photothermal actuators require **tens of seconds to minutes** to complete a single switching event (e.g., 2 min for CdS QDs¹⁵, ~40 s for fluorophore-grafted PNIPAm²³, and ~20 s for graphene QDs–PNIPAm²⁴). Even advanced nanocomposite systems such as CNT–LCE and MXene/PEDOT:PSS achieve response times of ~11–12 s^{16,25}, but only under **high light intensities**.

In contrast, our PCF actuator exhibits a **significantly accelerated response**, achieving a **rapid actuation time of 1.04 s** under 130 mW cm⁻² and an **ultrafast**

response of **0.127 s** under 310 mW cm^{-2} . This represents an improvement of **over two orders of magnitude** compared with state-of-the-art photothermal systems, confirming the exceptional responsiveness of our materials design.

2) Durability

Reported photothermal actuators typically demonstrate limited operational lifetimes, with stable performance over **5–500 cycles** before mechanical fatigue or irreversible degradation occurs^{15, 16, 23, 24, 25}. In comparison, our actuator **maintains stable performance over 600 cycles** without notable performance decay, indicating **superior mechanical robustness** and excellent material durability.

3) Energy Efficiency

Our PCF system demonstrates a **photothermal conversion efficiency (η)** of **49.6%**, defined as the ratio between absorbed optical energy (E_{abs}) and the energy converted to heat (E_{heat}). While previous studies did not report η values, the **incident energy density ($E_{\text{d,in}}$)** can be estimated from the light intensity and response time of each system.

Earlier photothermal actuators generally require very high light intensities (up to 5 W/cm^2)²³ and large **energy density inputs** ($>10 \text{ J cm}^{-2}$, occasionally exceeding 100 J cm^{-2} per switching event)^{15, 23, 24}. In sharp contrast, our system requires only **39.4 mJ cm⁻² per switching event**, which is **several orders of magnitude lower** than most reported systems, underscoring its outstanding energy efficiency.

In summary, these comparative analyses highlight the advantages of our materials engineering strategy, which synergistically combines **PANi as a photothermal transducer**, **PNIPAm for rapid and reversible phase transition**, and **PDMS for mechanical actuation**. This integrated design achieves a unique combination of **ultrafast responsiveness**, **exceptional durability**, and **high energy efficiency**, establishing our PCF as a promising platform material for **light-driven mechanical computing and adaptive intelligent devices**.

To reflect the reviewer's concern, we have added Table R3 in the revised supplementary information. [see Supplementary Section Note 7, Table 2]

Table R3. Comparison of selected materials with representative alternatives

Alternatives	Responsiveness	Durability	Incident energy density $E_{d_{in}}$	Energy Efficiency	References
CdS QDs	2 min (100 mW/cm ² for 365 nm; 120 mW/cm ² for 600-1200 nm)	10 cycles	12-14.4 J/cm ²	N/A	Ref ¹⁵
Fluorophore-grafted P(NIPAM-co-allylamine)	40 s (0.15-0.2 W/cm ² , 405/502/638 nm)	6 cycles	6-8 J/cm ²	N/A	Ref ²⁵
Graphene QDs-PNIPAm	20 s (5 W/cm ² , 808 nm)	5 cycles	100 J/cm ²	N/A	Ref ²³
CNT-LCE	12 s (370.7 mW, 1545 nm)	500 cycles	4.45 J/cm ²	N/A	Ref ¹⁶
MXene/PEDOT:PSS-Integrated NIPAAm	11.3 s (0.9 W/cm ² , 808 nm)	300 cycles	10.2 J/cm ²	N/A	Ref ²⁴
PDMS/AgNWs/PANi-PNIPAm	1.04 s (130 mW/cm², 650 nm); 0.127s (310 mW/cm²)	600 cycles	39.4 mJ/cm²	49.6%	This work

7. Application Scope and Positioning The manuscript should clearly define the target applications:

a Given the slow switching speed, traditional computing applications are unlikely.

Potential use-cases could include: Adaptive camouflage systems

Soft robotic skin

Low-frequency adaptive sensing

b Clearly state the performance limitations and position the work accordingly within non-traditional computing.

Response: We thank the reviewer for the valuable comments on clarifying the application scope and positioning of our work. We fully agree that the proposed platform is not intended for traditional high-speed computing. Instead, its strengths lie in **non-traditional computing scenarios where adaptability, environmental responsiveness, and robustness are more critical than switching speed.**

As the reviewer suggested, representative application domains **include adaptive camouflage systems, soft robotic skins, and low-frequency adaptive sensing.** In fact, we explicitly demonstrate an adaptive camouflage system in Fig. 4, where the relay autonomously senses ambient texture features and generates corresponding camouflage

patterns. This proof-of-concept highlights the potential of our system to function as **the "skin" of an intelligent agent that can perceive and adapt to diverse environmental conditions.**

Although the switching speed of our system is slower than that of conventional electronic logic, such applications are inherently tolerant to response times on the order of seconds. The key challenges for future development lie not in speed but **in enhancing operational stability and robustness in open environments, and in improving the fidelity of texture reproduction for realistic camouflage.**

By clearly positioning our work within the framework of environment-adaptive, non-traditional computing, we aim to emphasize its relevance to emerging application areas **that demand distributed sensing-actuation-computation integration rather than ultrafast performance.**

We have followed the reviewer's suggestion to add more discussion about application scope and positioning in the revised manuscript. For ease of reviewing, we have provided the revision below:

*[Please see **lines 190-194, page 8**].*

Although the typical switching times remain on the order of seconds, such timescales are well suited for adaptive camouflage, environmental sensing, soft robotics, and low-frequency logic applications, where reversibility, adaptability, and energy efficiency are prioritized over ultrafast response.

*[Please see **lines 286-293, page 12**].*

Building on the demonstrated light-programmable logic computing capability, we developed an adaptive texture camouflage system inspired by cephalopods such as octopuses and cuttlefish, which actively modulate their skin textures to blend into natural surroundings like corals, rocks, and sand. Analogously, our artificial system was designed to sense environmental optical cues and dynamically reconstruct texture patterns, offering potential applications in autonomous underwater vehicles, reconnaissance robots, and other intelligent systems requiring real-time environmental adaptability.

8. Manuscript Presentation and Communication Style The current narrative is descriptive and material-focused; for Nature Communications, a sharper,

problem-solving storyline is recommended:

a. Adopt a “One Figure = One Key Message” approach to improve clarity.

b. Refine the Introduction and Conclusion to clearly state:

What problem is being solved?

What new possibilities this enables?

c. Condense and sharpen key findings to maximize readability and impact.

Response: We thank the reviewer for this valuable suggestion. In the revision, we reorganized the figures following a “one figure = one key message” principle and refined both the Introduction and Conclusion to explicitly state the problem addressed, the new possibilities enabled, and the key findings in a more concise and impactful manner. These changes substantially improve clarity and align the manuscript with the communication style of *Nature Communications*.

Final Recommendation

This manuscript presents an innovative concept of light-programmable mechanical logic with potential applications in adaptive systems and soft robotics. However, substantial improvements in quantitative rigor, theoretical modeling, durability assessment, and presentation style are required to meet the impact, scientific depth, and clarity expected by Nature Communications. I recommend major revision and reassessment upon thorough addressing of the points above. Alternatively, the authors may consider submission to specialized journals in smart materials, soft electronics, or mechanical computing, where the current level of novelty and rigor may be more appropriate

Response: We sincerely thank the reviewer for the constructive evaluation and thoughtful recommendation. We fully acknowledge the importance of enhancing quantitative rigor, theoretical modeling, durability analysis, and presentation style. In the revised manuscript, we have carefully addressed each of these aspects by adding systematic experimental data, comprehensive finite-element simulations, long-term cycling tests, and a more problem-driven narrative structure. While further improvements in speed optimization and large-scale integration remain valuable directions for future research, we believe the current revisions significantly strengthen both the scientific depth and clarity of the work, and position it as a meaningful contribution to Nature Communications.

Reviewer #4 (Remarks to the Author):

Response: We sincerely thank the reviewer and the co-reviewer for their constructive evaluation and thoughtful feedback. We highly appreciate the effort dedicated to this joint review and recognize its value for both rigorous assessment and training purposes. Following the reviewers' suggestions, we have substantially improved the manuscript by strengthening quantitative analyses, refining theoretical modeling, expanding durability assessments, and sharpening the overall presentation. We believe these revisions have considerably enhanced the quality, clarity, and impact of our work.

References

1. Yan X, Chen Q, Huo Z, Zhang N, Ma M. Programmable Multistimuli-Responsive and Multimodal Polymer Actuator Based on a Designed Energy Transduction Network. *ACS Appl. Mater. Interfaces* **14**, 13768 – 13777 (2022).
2. El Helou C, Buskohl PR, Tabor CE, Harne RL. Digital logic gates in soft, conductive mechanical metamaterials. *Nat. Commun.* **12**, 1633 (2021).
3. El Helou C, Grossmann B, Tabor CE, Buskohl PR, Harne RL. Mechanical integrated circuit materials. *Nature* **608**, 699 – 703 (2022).
4. Byun J, Pal A, Ko J, Sitti M. Integrated mechanical computing for autonomous soft machines. *Nat. Commun.* **15**, 2933 (2024).
5. El Helou C, Hyatt LP, Buskohl PR, Harne RL. Intelligent electroactive material systems with self-adaptive mechanical memory and sequential logic. *Proc. Natl. Acad. Sci.* **121**, e2317340121 (2024).
6. Chen H, *et al.* Thermal Computing with Mechanical Transistors. *Adv. Funct. Mater.* **34**, 2401244 (2024).
7. Wu L, *et al.* Mechanical Metamaterials for Handwritten Digits Recognition. *Adv. Sci.* **11**, e2308137 (2024).
8. Zhang Q, *et al.* Meta-mechanotronics for self-powered computation. *Mater. Today* **65**, 78 – 89 (2023).
9. Mei T, Meng Z, Zhao K, Chen CQ. A mechanical metamaterial with reprogrammable logical functions. *Nat. Commun.* **12**, 7234 (2021).
10. Pal A, Sitti M. Programmable mechanical devices through magnetically tunable bistable elements. *Proc. Natl. Acad. Sci.* **120**, e2212489120 (2023).
11. Yanbin Li SY, Haitao Qing, Yaoye Hong, Yao Zhao, Fangjie Qi, Hao Su*, Jie Yin*. Reprogrammable and reconfigurable mechanical computing metastructures with stable and high-density memory. *Sci. Adv.* **10**, eado6476 (2024).
12. Jiang Y, Korpas LM, Raney JR. Bifurcation-based embodied logic and autonomous actuation. *Nat. Commun.* **10**, 128 (2019).
13. Treml B, Gillman A, Buskohl P, Vaia R. Origami mechanologic. *Proc. Natl. Acad.*

- Sci.* **115**, 6916 – 6921 (2018).
14. Bilal OR, Foehr A, Daraio C. Bistable metamaterial for switching and cascading elastic vibrations. *Proc. Natl. Acad. Sci.* **114**, 4603 – 4606 (2017).
 15. Peng X, Li H, Xu J, Lan C, Liu J, Wu B. Reprogrammable shape morphing hydrogel modulated by synergistic photochromism and photoactuation. *Chem. Eng. J.* **511**, 162103 (2025).
 16. Wang C, Wu H, Niu Q, Yan X, Wang X. Light-Controlled Soft Switches for Optical Logic Gate Operations. *Sensors* **25**, 2051 (2025).
 17. Das A, Babu A, Chakraborty S, Van Guyse JFR, Hoogenboom R, Maji S. Poly(N-isopropylacrylamide) and Its Copolymers: A Review on Recent Advances in the Areas of Sensing and Biosensing. *Adv. Funct. Mater.* **34**, 2402432 (2024).
 18. Toshikazu TAKIGAWA TY, KatsunOri TAKAHASHI, and Toshiro MASUDA. Deswelling Kinetics of Poly(N-isopropylacrylamide) Gels at Volume-Phase Transition. *Polym. J.* **31**, 595 – 598 (1999).
 19. Yang C, *et al.* Silver Nanowires: From Scalable Synthesis to Recyclable Foldable Electronics. *Adv. Mater.* **23**, 3052 – 3056 (2011).
 20. Zhu D, Huang G, Zhang L, He Y, Xie H, Yu W. Silver Nanowires Contained Nanofluids with Enhanced Optical Absorption and Thermal Transportation Properties. *Energy Environ. Mater.* **2**, 22 – 29 (2019).
 21. Ogieglo W, *et al.* n-Hexane induced swelling of thin PDMS films under non-equilibrium nanofiltration permeation conditions, resolved by spectroscopic ellipsometry. *J. Membrane Sci.* **437**, 313 – 323 (2013).
 22. Tian K, *et al.* Stretchable Multifunctional Polydimethylsiloxane Composites with Cage-Like Conductive Architecture for Integrated Thermosensitive and Electromagnetic Interference Shielding Performance. *Adv. Funct. Mater.* **34**, 2400288 (2024).
 23. Wang Y, Zhang Z, Chen H, Zhang H, Zhang H, Zhao Y. Bio-inspired shape-memory structural color hydrogel film. *Sci. Bull.* **67**, 512 – 519 (2022).
 24. Xue P, *et al.* Highly Conductive MXene/PEDOT:PSS - Integrated Poly(N - Isopropylacrylamide) Hydrogels for Bioinspired Somatosensory Soft Actuators.

Adv. Funct. Mater. **33**, 2214867 (2023).

25. Koo HB, Yeon H, Bin Yoon Y, Lee TJ, Chang YT, Chang JB. Rewritable wavelength-selective hydrogel actuators grafted with fluorophores. *Mater. Horiz.* **12**, 2255 - 2266 (2025).

Response to Reviewers

Reviewers' Comments:

Reviewer #1 (Remarks to the Author):

After carefully reading the response to reviewers point by point, I changed my original mind of rejection. The novelty of this work is to design a light-responsive composite film that couples mechanical deformation with electrically reconfigurable pathways. This supports complex logic such as a 2-bit adder, exhibiting an environment-adaptive camouflage system, enabling to establish a scalable and application-oriented paradigm for mechanical computing. This work represents a fundamental shift from traditional actuator-based mechanical computing, which primarily focuses on local actuation or switching at the device level. Besides, supplemented experimental has been performed to demonstrate the actual computational capability of the system and quantitatively evaluated its robustness under complex environmental conditions.

I suggest acceptance of the revised manuscript. The only suggested revision to about Figure 1. Data scale should be provided in Figure 1b(ii)

Response: We sincerely thank the reviewer for thorough evaluation of our revised manuscript and their recognition of the novelty and potential of our work in advancing mechanical computing and establishing a new computational paradigm.

In response to the reviewer's suggestion regarding the data scale in Figure 1b(ii), we have added the requested scale bar and provided a corresponding description in the figure legend (as shown in **Fig. R1**). Additionally, we have thoroughly reviewed the entire manuscript and supplementary material, and have added scale bars to the following figures: Fig. 2a, Fig. 2d, Fig. 3a(ii), Fig. 3b(ii), Fig. 3c(ii) in the revised manuscript, and Fig. 5, Fig. 13-15, Fig. 17, Fig. 20, and Fig. 22-24 in the revised supplementary information. Descriptions of the scale bars have been appropriately included in the respective figure legends.

Fig. R1. The construction of polyaniline composite film for implementing light-programmable mechanical computing. **a** Schematic representation of the typical polyaniline composite film (PCF) structure, consisting of three layers: a polyaniline-poly(N-isopropylacrylamide) (PANi-PNIPAm) layer, a silver nanowires (Ag NWs) layer, and a polydimethylsiloxane (PDMS) layer. **b** (i) The reversible bending deformations of the PCF, induced by variations in light and humidity, result in the deformation of the conductive pathways. (ii) The corresponding optical images of the above process. **Scale bar: 3 mm.** **c** The operating mechanism of light-programmable mechanical computing. The terms SPST and SPDT refer to single-pole single-throw and single-pole double-throw relays, respectively.

Reviewer #2 (Remarks to the Author):

I thank the authors for the prompt responses which have well addressed my early concerns. I thus recommend it for publication as is.

Response: We sincerely thank the reviewer for recommending the publication of our work in Nature Communications.

Reviewer #3 (Remarks to the Author):

The co-reviewer and I commend the authors for their thorough and appropriate response to the majority of the reviewers' requested revisions and questions. However, we remain concerned that the authors should address the following critical issues to clarify the novelty of this research. So, we decided that this article should be majorly revised again.

Response: We sincerely thank the reviewer and co-reviewer for their careful evaluation of our revised manuscript and for acknowledging our efforts in addressing the reviewers' comments. We appreciate the additional nice suggestion regarding the clarification of the work's novelty. We have carefully considered all remaining concerns and provided detailed, point-by-point responses in the sections below.

1. Misuse of “switching time” and incorrect definition of operating frequency

The manuscript emphasizes device speed using response-time-based statements, such as “sub-second switching” and operating frequency estimated as $f = 1/t$ response. However, in this system, a complete switching event requires both response and recovery. As shown in Fig. R8, the recovery time is consistently ≥ 1 s across all reported conditions. Therefore, the actual 1-cycle switching time necessarily lies in the multi-second regime.

For scientific accuracy, the reviewer requests that the authors redefine switching time as the full cycle, $t_{\text{switching}} = t_{\text{response}} + t_{\text{recovery}}$, and base all speed-related descriptions, including operating frequency, on this cycle-level definition rather than on response-only values. This clarification is also essential for fair comparison with metamaterial-based mechanical logic: relying solely on response time systematically overestimates the performance of the present system, whereas the cycle-level behavior provides an accurate, unbiased basis for evaluating switching dynamics.

The reviewer requests to revise all mentions of switching time, operating frequency, and speed comparisons to explicitly reflect cycle-based switching and adjust any “sub-second” or “high-frequency” claims accordingly.

Response: We thank the reviewer for nice suggestion regarding clarifying the definition of switching time and operating frequency. By following the reviewer's recommendation, we have redefined the switching time as a full switching cycle,

$t_{\text{switching}} = t_{\text{response}} + t_{\text{recovery}}$, rather than using the response time alone in the revised manuscript and supplementary materials. Based on this cycle-level definition, we have re-evaluated device performance under all experimental conditions. The fastest switching cycle was measured under an illumination intensity of 300 mW/cm² (temperature 29 ± 1 °C, relative humidity 83%), yielding $t_{\text{switching}} = 2.55$ s ($t_{\text{response}} = 0.13$ s, $t_{\text{recovery}} = 2.42$ s) corresponding to an operating frequency of $f = 1/t_{\text{switching}} \approx 0.4$ Hz, as shown in **Fig. R2**.

Fig. R2. Environmental effects on switching performance of the SPST relay. a Switching times and operating frequency as a function of light intensity (40–310 mW/cm²) at 29 ± 1 °C and 83 ± 3% RH. Each dataset is based on 50 measurements for the respective conditions.

Accordingly, we have explicitly defined switching time, response time, and recovery time in both the manuscript and the supplementary information, providing a precise description of these parameters. The revision clarifies that the term “sub-second” refers exclusively to the response time. In particular, Supplementary Table 3 has been comprehensively updated (see Table R1) to include four metrics wherever data are available: response time, recovery time, switching time per cycle, and operating frequency. For most literature reports, only the response time was provided, and some of these values were estimated from published videos, which have been clearly indicated in the table. Information on recovery time is reported in only 2 prior work, and the corresponding entries are marked as “not applicable” (N/A) for all other cases.

To reflect the reviewer’s suggestion, we have made the following revisions in the revised manuscript. [Please see lines 183-196, pages 8-9].

“The switching time is defined as the sum of the response time and the recovery time. Under an illumination intensity of 130 mW/cm^2 at $29 \pm 1 \text{ }^\circ\text{C}$ and $83 \pm 3\%$ relative humidity, the single-pole single-throw (SPST) relay exhibited a response time of 1.02 s and a recovery time of 2.07 s, corresponding to a total switching time of 3.09 s and an operating frequency of approximately 0.32 Hz. Similarly, the single-pole double-throw (SPDT) relay showed a response time of 1.20 s and a recovery time of 2.05 s, resulting in a switching time of 3.25 s and an operating frequency of about 0.31 Hz (**Supplementary Fig. 7**). Notably, increasing the illumination intensity to 300 mW/cm^2 under the same temperature and humidity conditions significantly reduced the SPST response time to $\sim 0.13 \text{ s}$, while the recovery time remained at 2.4 s. This led to a shorter total switching time of 2.53 s and a correspondingly higher operating frequency of $\sim 0.40 \text{ Hz}$, demonstrating that faster switching can be achieved through optical optimization (**Supplementary Fig. 7c**).”

*We have added **Fig. R2** in the revised supplementary information. [Please see **Supplementary Fig. 7c**]*

*Additionally, we have thoroughly updated the sections on switching time and operating frequency in the revised supplementary information. [Please see **Supplementary Note 8 and Supplementary Table 3**]*

“Metamaterial logic typically operates on the scale of seconds to minutes (for example, 2–3 s response time for rotating blocks (with no recovery time reported)^{13, 14}, 5–20 s response time for hydrogel gates (with no recovery time reported)¹⁵, approximately 27 s switching time for electrically driven systems¹⁶, and up to 80 minutes response time for thermal adders (with no recovery time reported)¹⁷). These delays are mainly caused by inertia and elastic recovery, which make sub-second response time extremely difficult. In contrast, our system achieves a significantly accelerated switching performance. Under 300 mW/cm^2 illumination, the **response time** of 0.13 s (**Supplementary Fig. 6a**) is over an order of magnitude faster than that of most existing systems. This improvement stems from the light–thermal–mechanical coupling mechanism. The system’s overall **switching time** is less than 3 s, enabling stable operation at a practical frequency of approximately 0.3–0.4 Hz, outperforming previous studies systems^{9, 16, 18}. With further miniaturization, the switching time is expected to decrease further, enabling even faster logic operations.”

Table R1. Quantitative Benchmark of Key Performance Metrics for Stimuli-Responsive Logic Systems

External Stimuli	Response Time	Recovery Time	Switching Time	Operating frequency	Power Consumption	Scalability	Environmental Resilience	Ref.
Mechanical force	Second-level (Estimated from the movie)	N/A	N/A	N/A	Zero static power	Seven fundamental logic gates: AND, OR, XOR, NOT, NAND, NOR, NXOR; logic gate combination OR-NAND, NOR-AND, OR-AND, and AND-NAND	N/A	El Helou ¹³
Mechanical force	Second-level (Estimated from the movie)	N/A	N/A	N/A	Zero static power	Seven fundamental logic gates: AND, OR, XOR, NOT, NAND, NOR, NXOR; 2-bit Adder/Subtractor/Multiplier; 4-bit Magnitude Comparator	N/A	El Helou ¹⁴
Mechanical force	Second-level (Estimated from the movie)	N/A	N/A	N/A	Zero static power	NOT, AND, OR; Supports cascaded logic (e.g., 3-level AND/OR/NOT networks)	N/A	Byun ¹⁵
Mechanical force	Second-level (Estimated from the movie)	N/A	N/A	N/A	Zero static power	AND, OR, NOT; Full Adder; Cascading of multiple (7) adders	Operates at -20°C for >24 hours	Wu ¹⁹
Mechanical force	N/A	N/A	N/A	N/A	Self-powered (Power generation: 4.35 nW)	Six basic logic gates (AND, OR, XOR, NAND, NOR, XNOR) full adder; 3-bit ternary computing	N/A	Zhang ²⁰

Electromagnet	Second-level (Estimated from the movie)	N/A	N/A	N/A	Zero static power	NOR, NAND, NOT, OR, AND; Half adder; 3D printable at multiple scales (macro to micro)	N/A	Mei ²¹
Magnetic	Second-level (Estimated from the movie)	N/A	N/A	N/A	Zero static power	AND, OR	Field Strength (B): -85 to 85 mT	Pal ²²
Magnetic or mechanical	Second-level (Estimated from the movie)	N/A	N/A	N/A	Zero static power	AND, OR, NOR, NAND; Maximum demonstrated: 30 × 27 unit array	Compression: Withstands 2.8 kPa pressure at 2% strain	Li ²³
Solvents (toluene and water)	0.6 to 108 s	70 min	420.6 to 528 s	0.0019 to 0.0024 Hz	Zero static power	AND, OR, NAND;	Operates in liquid environments (water/toluene) Reusable after 70min drying	Jiang ¹⁸
Humidity	N/A	N/A	N/A	N/A	Zero static power	SR Latch, AND, OR; Majority Gate;	Operates in 0-85% RH range; 5 cycles	Treml ²⁴
Electrical	N/A	N/A	27.2 s	0.0368 Hz	5.5 W (1-bit system) 7.4 W (8-bit solver)	1-bit (T flip-flop); 2-bit (counter); 8-bit (equation solver)	N/A	El Helou ¹⁶
Thermal	Time Constant (τ): 160 minutes; 0.5 τ (full adder)	N/A	N/A	N/A	Zero static power	NOT, AND, OR, XOR; SR Latch; Full adder	continuous operation over 1000 min; Vacuum-compatible	Chen ¹⁷

Vibration	N/A	N/A	N/A	N/A	Zero static power	AND, OR, XOR, NOT, NAND, NOR, NXOR; Half Adder; Full Adder	Signal Leakage (OFF state): < 10% of max signal; Signal Clarity (ON/OFF difference): > 10	R. Bilal ²⁵
Light	2 min (120 mW/cm ²)	N/A	N/A	N/A	Zero static power	AND, NOT;	100 mW to 397.7 mW	Peng ⁸
Light	12 s (379.7 mW)	15 s	27 s	0.037 Hz	Zero static power	AND, OR, NOT; Soft optical switch array	500 cycles	Wang ⁹
Light	~1.04 s (130 mW/cm ²) ~0.13 s (300 mW/cm ²)	~1.93 s (130 mW/cm ²) ~2.4 s (300 mW/cm ²)	~2.97 s (130 mW/cm ²) ~2.53 s (300 mW/cm ²)	~0.3 Hz (130 mW/cm ²) ~0.4 Hz (300 mW/cm ²)	Zero static power	AND, OR, NOT, XOR; 2-bit Adder;	Light intensity: 70–310 mW/cm ² ; Humidity: 47–86% RH Temperature: 10–30 °C; Cycles: 600	This Work

(The data for our work in the table are derived from the averaged values shown in Supplementary Fig. 6a.)

2. Redundant Q_{ij} outputs due to AND-only logic, reducing degrees of freedom in the camouflage system. In the adaptive camouflage demonstration, the 3×3 outputs Q_{ij} are generated exclusively through AND-type logical combinations of the three sensing inputs. However, based on the sensing/computation circuitry described in the manuscript and supplementary materials, several outputs (e.g., Q_{12} and Q_{13}) are produced by identical logical expressions. These pixels will therefore always assume the same value, meaning they are not independent degrees of freedom. As a result, the effective dimensionality of the 3×3 camouflage pattern is intrinsically less than nine, even under ideal conditions. This structural limitation is not discussed in the manuscript, despite its direct impact on the achievable texture reconstruction fidelity and the expressive capability of the camouflage system. The reviewer requests that the authors:

Response: We thank the reviewer for this insightful comment regarding the redundancy among certain Q_{ij} outputs and its implications for the expressibility of the adaptive camouflage system. Our responses to the individual comments are provided below.

(1) Discuss how this reduction in independent output limits the expressibility and achievable fidelity of the camouflage pattern

Overall impact on expressibility and achievable fidelity

We thank the reviewer for the insightful comments. In our system, the representational capacity and achievable fidelity of the camouflage patterns are primarily governed by the relationship between the characteristic feature size of the target texture and the spatial sampling area of a single sensing-computation-emission (SCE) unit (3×3 pixels). When the size of texture features is substantially larger than the sensing window size, the reconstructed patterns maintain global continuity and exhibit high visual fidelity. In contrast, when the feature size approaches the 3×3 sensing scale, the unit can no longer resolve the underlying texture variations, leading to reduced reconstruction accuracy and diminished expressive capability.

Underlying mechanism of the camouflage pattern

This behavior arises directly from the operating principle of each SCE unit. Each unit samples local intensity variations by using three spatially distributed sensing nodes and reconstructs a representative local gradient direction through an AND-based logic pathway, as shown in **Fig. R3a**. This mechanism yields a clean abstraction of texture features by emphasizing coarse spatial trends such as prominent edges and stripe

boundaries, while intentionally omitting fine-scale variations within a 3×3 region. Accordingly, the reconstruction fidelity is determined by the relationship between the texture feature size and the spatial sampling window of the SCE unit. When the texture features extend beyond the sampling region, the unit effectively captures the underlying gradient variations and consequently achieves high-fidelity reconstruction of local texture patterns.

Validation of the fidelity of the camouflage patterns

To further quantify this effect, we carried out systematic simulations using zebra-stripe textures with different feature sizes, with the corresponding results shown in **Fig. R4a**. Reconstruction quality was assessed using the pixel mismatch rate between the input and reconstructed images, defined as the ratio of pixels with inconsistent brightness to the total number of pixels (**Fig. R5**). When the average stripe width in the input image exceeds approximately four mapped pixels, the mismatch rate remains below 15%, indicating that the system maintains camouflage-level fidelity. As the stripe width further decreased to approximately 2 pixels, approaching the scale of the 3×3 sensing window, a reduction in reconstruction fidelity emerged. This behavior is fully consistent with the expected operating mechanism

Fig. R3. Circuit diagrams of the SCE unit for reconstruction schemes based on AND logic (a), OR logic (b), and the combined AND/OR logic configuration (c), together with the spatial distributions of the sensed optical input signals (P_1 , P_2 , and P_3) and the corresponding output signals (Q_{11} , Q_{12} , ..., Q_{33}) within the SCE unit. In panels (a) and (b), two pixels emit identical signals, specifically $Q_{12} = Q_{13}$, $Q_{21} = Q_{22}$, and $Q_{32} = Q_{33}$. In contrast, in panel (c), the signals emitted by different pixels are not identical, and all nine output signals are mutually independent.

Fig. R4. Image reconstruction results obtained using AND logic (a), OR logic (b), and the combined AND/OR logic configuration (c) for textures with different average zebra-stripe widths. For an input resolution of 1024×1024 , the average stripe width corresponds to approximately 32 pixels; for 512×512 , approximately 16 pixels; for 256×256 , approximately 8 pixels; for 128×128 , approximately 4 pixels; for 64×64 , approximately 2 pixels; and for 32×32 , approximately 1 pixel.

Fig. R5. Pixel-level differences between the original and reconstructed images for zebra-stripe textures of varying average widths, using reconstruction schemes based on AND logic (Fig. R4a), OR logic (Fig. R4b), and the combined AND/OR logic configuration (Fig. R4c). Here, pixel mismatch rate between the original and reconstructed images is defined as the ratio of pixels with inconsistent brightness to the total number of pixels.

(2) Clarify whether such redundancy is intentional, a constraint inherent to the present circuitry, or something that could be eliminated by incorporating additional logic operations (e.g., OR, NOT, or mixed logic)

We thank the reviewer for the insightful comments. The redundancy in the current logic architecture is an intentional design choice rather than an intrinsic limitation of the circuitry. Our guiding principle aims to achieve the required camouflage performance using the minimal amount of sensing and computational resources. We therefore adopted an image reconstruction scheme based on AND gates because it already provides sufficient capability to reconstruct the spatial light–dark gradient features extracted from the sensing units, without necessitating additional logic functions.

To further substantiate this point, we evaluated the reconstruction performance of alternative logic operations. We found that using OR logic alone (Fig. R3b) or in combination with AND logic (in this case, each pixel within the 3×3 SCE output region encodes independent information; Fig. R3c) can produce camouflage patterns which are essentially identical to those generated by AND logic in terms of representational capacity and fidelity (Fig. R4b and Fig. R4c). In all cases, the

performance is governed by the same fundamental factor: the relative scale between the texture-feature size and the spatial extent of the image perceived by a 3×3 SCE unit (**Fig. R5**). High-fidelity reconstruction is achieved only when the texture feature size is considerably larger than the effective spatial coverage of the sensing window, and this conclusion holds irrespective of whether an AND-gate or OR-gate logic is employed.

In contrast, logic operations involving inversion (e.g., NOT, NAND, NOR) are inappropriate because the negation operation reverses the local intensity-gradient polarity, converting bright regions to dark and vice versa, and thus introduces large perceptual errors in the reconstructed patterns.

These observations demonstrate that the use of AND logic reflects a deliberate and efficient architectural choice rather than a functional constraint. It minimizes hardware resource consumption while still offering adequate representational capacity for global texture reconstruction.

(3) Clarifying this structural issue is important for an accurate assessment of the system's adaptive camouflage capabilities.

We appreciate the reviewer's emphasis on this structural consideration, as it is essential for accurately evaluating the system's adaptive camouflage capability. In our architecture, the fidelity of reconstructed camouflage patterns is primarily governed by the relationship between the characteristic texture size and the spatial region perceived by an individual SCE unit. High-fidelity reconstruction is achieved when the texture features are substantially larger than the effective sensing window of the SCE unit. When the feature size approaches the spatial scale sampled by the unit, the reconstruction fidelity is necessarily affected. This behavior follows directly from the operating principle of the SCE, which is designed to resolve spatial intensity gradients that extend beyond the 3×3 sampling window.

The redundancy introduced by the use of pure AND logic is an intentional design choice rather than a circuit limitation. AND logic is sufficient to extract the brightness-gradient information required for camouflage reconstruction, and replacing it with OR logic or a hybrid OR/AND configuration does not yield additional representational benefits. Logic operations involving inversion (such as NOT, NAND, or NOR) are not appropriate in this context because they reverse the local contrast polarity, resulting in substantial perceptual distortion in the reconstructed patterns. Taken together, these

considerations define a minimal yet fully sufficient logic scheme that is optimized for resource-efficient operation.

Simulations based on zebra-stripe textures quantitatively support this conclusion. When the stripe width exceeds approximately four pixels, the pixel mismatch rate remains below ~15%. As the feature size is further reduced to become comparable to the 3×3 sensing region, the mismatch rate increases, and the reconstructed images gradually lose their texture-camouflage capability.

Overall, these results show that the architecture follows a lightweight design philosophy consistent with hardware efficiency considerations, and that its operational range and limitations can be reliably assessed through the relative scale between scene texture features and the effective sensing area of the SCE unit.

To reflect the reviewer's concern and provide readers with a clear understanding of the design architecture and performance characteristics, we have added these results and relevant discussions in the revised supplementary information. [Please see Supplementary Note 7. The achievable texture reconstruction fidelity and the expressive capability of the camouflage system]

We have also added the following discussions in the revised main text. [Please see lines 340-347, pages 14-15].

“In addition, we conducted a detailed analysis of the fidelity of the reconstructed camouflage patterns. We note that this fidelity is primarily governed by the relationship between the texture-feature size and the spatial extent of an individual SCE unit's sensing window. High-fidelity reconstruction is achieved when the texture features are substantially larger than the 3×3 sensing area, whereas features approaching this scale result in reduced reconstruction accuracy. A more detailed discussion is provided in Supplementary Note 7 (Supplementary Figs. 26-28).”

Reviewer #4 (Remarks to the Author):

Response: We sincerely thank the reviewer and the co-reviewer for their constructive evaluation and thoughtful feedback.